# Exactly Tight Information-theoretic Generalization Bounds via Binary Jensen-Shannon Divergence

Yuxin Dong[1]  Haoran Guo[1]  Tieliang Gong[1]  Wen Wen[1]  Chen Li[1]

## Abstract

Information-theoretic bounds, while achieving significant success in analyzing the generalization of randomized learning algorithms, have been criticized for their slow convergence rates and overestimation. This paper presents novel bounds that bridge the expected empirical and population risks through a binarized variant of the Jensen-Shannon divergence. Leveraging our foundational lemma that characterizes the interaction between an arbitrary and a binary variable, we derive hypothesis-based bounds that enhance existing conditional mutual information bounds by reducing the number of conditioned samples from 2 to 1. We additionally establish prediction-based bounds that surpass prior bounds based on evaluated loss mutual information measures. Thereafter, through a new binarization technique for the evaluated loss variables, we obtain exactly tight generalization bounds broadly applicable to general randomized learning algorithms for any bounded loss functions. Our results effectively address key limitations of previous results in analyzing certain stochastic convex optimization problems, without requiring additional stability or compressibility assumptions about the learning algorithm.

## 1. Introduction

Characterizing the behavior of generalization error is a cornerstone of statistical learning theory. Traditional approaches grounded in complexity measures such as VC-dimension or Rademacher complexity often fail to capture the dynamics of modern iterative and noisy optimization algorithms, such as stochastic gradient descent (SGD) and Adam. Consequently, these methods typically yield overly

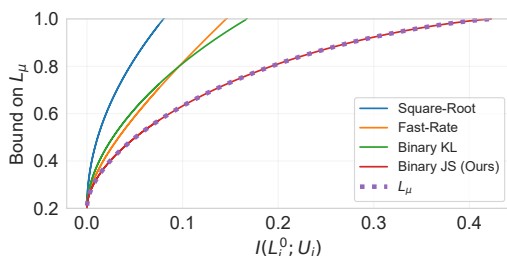

Figure 1: Visualization of information-theoretic generalization bounds for the population risk $L_\mu$ evaluated using 0-1 loss with empirical risk $L_n = 0.2$.

conservative or even vacuous generalization bounds.

Employing information-theoretic measures for analyzing generalization properties has gained considerable traction following the pioneering work of (Russo & Zou, 2019; Xu & Raginsky, 2017). This paradigm quantifies the amount of information about the training data encoded in model parameters. Unlike conventional techniques, these information-theoretic approaches deliver generalization bounds that are both data and algorithm-dependent with significantly weaker assumptions compared to those of algorithm stability (Hardt et al., 2016; Bassily et al., 2020) or model compression (Arora et al., 2018; Zhou et al., 2019). Recent advancements in this domain have substantially enhanced our understanding of stochastic gradient-based learning algorithms (Negrea et al., 2019; Neu et al., 2021; Wang & Mao, 2021).

Despite their appeal, information-theoretic bounds have faced substantial criticism for their suboptimal convergence rates. Specifically, when the key information quantities are bounded by a constant, these bounds typically decay at a rate of $O(\frac{1}{\sqrt{n}})$, which contrasts with the faster $O(\frac{1}{n})$ rate frequently observed in practical learning scenarios, e.g., convex or strongly-convex optimization problems. These bounds may be inherently suboptimal when faster rates are achievable. To mitigate this limitation, (Hellström & Durisi, 2021; Wang & Mao, 2023a) introduce fast-rate bounds by leveraging a linear combination of the empirical risk and information-theoretic generalization terms. Meanwhile, (Hellström & Durisi, 2022b) explores an unconventional relationship between the expected empirical and population risks using binary KL divergence. Although these ap-

[1]School of Computer Science and Technology, Xi'an Jiaotong University. Correspondence to: Tieliang Gong <adidasgtl@gmail.com>.

proaches yield faster decay rates when the empirical risk ($L_n$) approaches or equals zero, they still become vacuous for higher population risks ($L_\mu$), as illustrated in Figure 1.

Another prevalent critique of information-theoretic bounds pertains to their reliance on input-output mutual information, which often results in severe overestimation for modern deep neural networks. To address this, one line of research employs the individual-sample technique (Bu et al., 2020), enabling tighter information-theoretic generalization terms. This methodology has been extended to the super-sample generalization framework (Steinke & Zakynthinou, 2020), which facilitates tightened bounds through conditioning techniques (Haghifam et al., 2020; Zhou et al., 2022a). A complementary line of research refines bounds by reducing the dimensionality of the associated random variables, including network predictions (Harutyunyan et al., 2021), loss pairs (Hellström & Durisi, 2022b), or loss differences (Wang & Mao, 2023a). These bounds not only yield tighter estimates but also enhance computational tractability. However, all these bounds fail to vanish in at least one of the counterexamples demonstrated by (Haghifam et al., 2023), highlighting their inherent limitations.

In this paper, we propose a new approach to characterizing the relationship between expected empirical and population risks through a binarized variant of the Jensen-Shannon (JS) divergence, which achieves faster convergence compared to existing fast-rate and binary KL-based methods. Additionally, we introduce tighter information-theoretic generalization measures utilizing binarized loss variables, surpassing the tightness of current loss pair or loss difference approaches. By combining these techniques, we successfully derive exactly tight information-theoretic bounds for general randomized learning algorithms. The key contributions of this paper are listed as follows:

- We demonstrate that the interaction between an arbitrary variable and a binary variable can be effectively captured using the proposed binary JS divergence (Lemma 3.1), enabling us to derive new information-theoretic generalization bounds for the expected population risk.

- Motivated by recent work on hypothesis-based generalization bounds, which suggest that removing redundant variables from the conditional mutual information tightens the bounds, we establish a new bound (Theorem 3.2) by conditioning on a single sample instead of the previous sample pair approach as in (Zhou et al., 2022a), achieving tighter upper-bound estimates as well as exact tightness for some specific learning tasks.

- Inspired by the single-loss mutual information framework explored in (Wang & Mao, 2023a), we introduce a single-loss binary JS bound (Theorem 3.6), which is significantly tighter than existing square-root, fast-rate, and binary KL-based bounds. Our bound provides strictly

non-vacuous guarantees for the population risk and is exactly tight for binary loss functions.

- We propose a novel binarization technique for evaluated loss variables, leading to exactly tight generalization bounds (Corollary 3.10) for arbitrary bounded loss functions. Unlike prior works, which focus on exact characterizations for interpolating settings (Wang & Mao, 2023a) or specific categories of learning algorithms (Aminian et al., 2021; Zhou et al., 2024), our results are broadly applicable to general randomized learning algorithms without imposing additional strong assumptions.

- We extend our exactly tight results to accommodate unbounded loss functions (Corollary 3.12) and generalize beyond KL divergence to include $f$-divergence (Theorem 4.2) and Wasserstein distance (Corollary 4.3).

- We validate the effectiveness of our bounds in capturing generalization dynamics across diverse synthetic and real-world learning scenarios. The results demonstrate that our bounds consistently outperform existing methods and precisely track the true generalization error.

## 2. Preliminaries

Throughout this paper, random variables are denoted by capitalized letters ($X$), their specific realizations by lowercase letters ($x$), and the corresponding spaces by calligraphic letters ($\mathcal{X}$). The distribution of a variable $X$ is denoted by $P_X$, the conditional distribution of $X$ given $Y$ is $P_{X|Y}$, and the conditional distribution given a specific realization is $P_{X|Y=y}$. Expectations taken over $X \sim P_X$ are denoted by $\mathbb{E}_X$. We use $H(X)$ to represent Shannon's (differential) entropy, and $D_{\mathrm{KL}}(P \| Q)$ to denote the Kullback-Leibler (KL) divergence of $P$ relative to $Q$. The mutual information between variables $X$ and $Y$ is denoted by $I(X; Y)$, and their conditional mutual information given $Z$ is $I(X; Y|Z)$. Additionally, we define the disintegrated mutual information as $I^z(X; Y) = D(P_{X,Y|z} \| P_{X|z} P_{Y|z})$. The logarithmic function $\log$ is assumed to have base $e$.

### 2.1. Generalization Error

Let $\mathcal{Z} = \mathcal{X} \times \mathcal{Y}$ denote the instance space, where $\mathcal{X}$ and $\mathcal{Y}$ represent the input and label spaces, respectively. The training dataset $\mathbf{Z} = \{Z_i\}_{i=1}^n \in \mathcal{Z}^n$ is constructed by i.i.d. sampling from the data-generating distribution $\mu$. A learning algorithm $\mathcal{A}$ takes $\mathbf{Z}$ as input and outputs a hypothesis $W = \mathcal{A}(\mathbf{Z}) \in \mathcal{W}$, characterized by the conditional distribution $P_{W|\mathbf{Z}}$. Let $\ell : \mathcal{W} \times \mathcal{Z} \to \mathbb{R}^+$ be the loss function. For a given $w \in \mathcal{W}$, the population risk is defined as $L_\mu(w) \triangleq \mathbb{E}_Z[\ell(w, Z)]$, where $Z \sim \mu$ is an independent test sample. The expected population risk is denoted by $L_\mu = \mathbb{E}_W[L_\mu(W)]$. Since the true distribution $\mu$ is generally unknown in practice, the empirical risk is used instead and is defined as $L_\mathbf{Z}(w) \triangleq$

$\frac{1}{n} \sum_{i=1}^{n} \ell(w, Z_i)$. Similarly, the expected empirical risk is $L_n = \mathbb{E}_{W,\mathbf{Z}}[L_{\mathbf{Z}}(W)]$. The generalization error is quantified as $\overline{\text{gen}}(W, \mathbf{Z}) \triangleq L_\mu(W) - L_{\mathbf{Z}}(W)$, measuring the discrepancy between the population and empirical risks. Under the average case, it is abbreviated as $\overline{\text{gen}} = \mathbb{E}_{W,\mathbf{Z}}[\overline{\text{gen}}(W, \mathbf{Z})]$.

## 2.2. Supersample Setting

The supersample framework, first introduced in (Steinke & Zakynthinou, 2020), provides a powerful approach for generalization analysis. Let $\widetilde{\mathbf{Z}} = \{\widetilde{Z}_i\}_{i=1}^{n} \in \mathcal{Z}^{n \times 2}$ denote a supersample dataset drawn i.i.d. from $\mu$, where each element $\widetilde{Z}_i = (\widetilde{Z}_i^0, \widetilde{Z}_i^1)$ consists of a pair of samples. A set of binary variables $U = \{U_i\}_{i=1}^{n} \sim \text{Unif}(\{0,1\}^n)$ is used to partition the supersample dataset into training and test datasets. Specifically, the training dataset is $\widetilde{\mathbf{Z}}_U = \{\widetilde{Z}_i^{U_i}\}_{i=1}^{n}$, while the test dataset is $\widetilde{\mathbf{Z}}_{\overline{U}} = \{\widetilde{Z}_i^{\overline{U}_i}\}_{i=1}^{n}$, where $\overline{U}_i = 1 - U_i$. The empirical and population risks in this setup are formulated as $L_n = \mathbb{E}_{W,\widetilde{\mathbf{Z}},U}[L_{\widetilde{\mathbf{Z}}_U}(W)]$ and $L_\mu = \mathbb{E}_{W,\widetilde{\mathbf{Z}},U}[L_{\widetilde{\mathbf{Z}}_{\overline{U}}}(W)]$ respectively. Additionally, let $L_i^u = \ell(W, \widetilde{Z}_i^u)$ for $u \in \{0,1\}$ denote the evaluated loss. A pair of losses is represented as $L_i = (L_i^0, L_i^1)$, and their difference is defined as $\Delta_i = L_i^1 - L_i^0$.

## 2.3. Binary KL Divergence

The work of (Hellström & Durisi, 2022b) introduces an approach to bound the binary KL divergence between the expected empirical and population risks, defined as follows:

$$d_{\text{KL}}(p \,\|\, q) \triangleq p \log\left(\frac{p}{q}\right) + (1-p) \log\left(\frac{1-p}{1-q}\right). \quad (1)$$

This framework is further refined in (Dong et al., 2024a) by eliminating the conditional dependence on $\widetilde{\mathbf{Z}}$. The binary KL divergence $d_{\text{KL}}(L_n \,\|\, L_\mu)$ delineates an unconventional relationship between $L_n$ and $L_\mu$, often yielding bounds that are tighter than the square-root or fast-rate counterparts proposed in (Harutyunyan et al., 2021). The proof hinges on a relaxed form of binary KL divergence, defined as:

$$d_\gamma(p \,\|\, q) \triangleq \gamma p - \log(1 - q + qe^\gamma).$$

A key property of this relaxed binary KL divergence is:

$$d_{\text{KL}}(p \,\|\, q) = \sup_\gamma d_\gamma(p \,\|\, q).$$

Notably, both $d_{\text{KL}}(\cdot \,\|\, \cdot)$ and $d_\gamma(\cdot \,\|\, \cdot)$ are jointly convex in their arguments. Additionally, $d_\gamma(\cdot \,\|\, \cdot)$ exhibits linearity w.r.t its first argument. These properties serve as foundational elements for the subsequent theoretical analysis.

# 3. Generalization Bounds via Binary Jensen-Shannon Divergence

In this section, we present a new family of generalization bounds based on the binary JS divergence between the ex-

pected empirical and population risks, defined as:

$$d_{\text{JS}}(p \,\|\, q) \triangleq \tfrac{1}{2} d_{\text{KL}}\left(p \,\|\, \tfrac{p+q}{2}\right) + \tfrac{1}{2} d_{\text{KL}}\left(q \,\|\, \tfrac{p+q}{2}\right). \quad (2)$$

As the JS divergence serves as a proper metric of distance in the space of probability distributions, the binary JS divergence can be interpreted as a quantifiable measure of the "distance" between empirical and population risks.

We begin our exploration by presenting a foundational lemma upon which the main results in this section are built. This inequality introduces a new perspective for characterizing the interaction between a $\frac{1}{2}$-Bernoulli variable and an arbitrary random variable. Notably, this result may hold significance beyond the context of generalization analysis, offering potential applications in broader aspects.

**Lemma 3.1.** *Given random variable $X$, binary variable $Y \sim \text{Bern}\left(\frac{1}{2}\right)$ and a measurable function $f(x)$. Assume that $f(X) \in [0,1]$ almost surely, then*

$$d_{\text{JS}}\left(\mathbb{E}_{X|Y=0}[f(X)] \,\big\|\, \mathbb{E}_{X|Y=1}[f(X)]\right) \leq I(X;Y).$$

*Additionally, if $f(x)$ is invertible and $f(X) \in \{0,1\}$ almost surely, the inequality above holds with equality.*

The proof, which is derived using Jensen's inequality and the Donsker-Varadhan formula for KL divergence, is provided in Appendix B along with proofs of all other results.

This lemma integrates seamlessly with the supersample framework to analyze the interaction between supersample variables $U$ and the hypothesis or predictions. For illustration, consider a supersample dataset containing two samples, $Z_0$ and $Z_1$, where $Z_0$ is used for training and $Z_1$ for testing. Define a loss variable evaluated as $L = f(W, Z_U)$, where $U \sim \text{Bern}\left(\frac{1}{2}\right)$ randomly selects $Z_U \in \{Z_0, Z_1\}$. It is straightforward to verify that $L_n = \mathbb{E}_{L|U=0}[L]$ and $L_\mu = \mathbb{E}_{L|U=1}[L]$. By setting $X = L$, $Y = U$, and $f(l) = l$ in Lemma 3.1, we establish the following bound:

$$d_{\text{JS}}(L_n \,\|\, L_\mu) \leq I(L; U). \quad (3)$$

Using the inverse of the binary JS divergence, defined as:

$$d_{\text{JS}}^{-1}(p, c) \triangleq \sup\{q \in [0,1] : d_{\text{JS}}(p \,\|\, q) \leq c\}, \quad (4)$$

we can now characterize the population risk $L_\mu$ in terms of the empirical risk $L_n$ and $I(L; U)$:

$$L_\mu \leq d_{\text{JS}}^{-1}(L_n, I(L; U)). \quad (5)$$

Next, we leverage Lemma 3.1 to explore several classical types of information-theoretic generalization bounds, demonstrating the substantial potential of this lemma.

## 3.1. A Hypothesis-based Generalization Bound

Consider the scenario where $X = W$, $Y = U_i$, and $f(w) = \ell(w, \widetilde{Z}_i^0)$ all conditioned on $\widetilde{Z}_i^0$. Building on Lemma 3.1, the following bound is established:

Table 1: Summary of hypothesis-based information-theoretic generalization approaches in the literature.

| Approach | Reference | Measure | Bound for $\ell(\cdot,\cdot) \in [0,1]$ |
|---|---|---|---|
| Mutual Information (MI) | (Xu & Raginsky, 2017) | $I(W;\mathbf{Z})$ | $\sqrt{\frac{1}{2n}I(W;\mathbf{Z})}$ |
| Individual MI (IMI) | (Bu et al., 2020) | $I(W;Z_i)$ | $\frac{1}{n}\sum_{i=1}^{n}\sqrt{\frac{1}{2}I(W;Z_i)}$ |
| Conditional MI (CMI) | (Steinke & Zakynthinou, 2020) | $I(W;U|\widetilde{\mathbf{Z}})$ | $\sqrt{\frac{2}{n}I(W;U|\widetilde{\mathbf{Z}})}$ |
| Conditional Individual MI (CIMI) | (Haghifam et al., 2020) | $I(W;U_i|\widetilde{\mathbf{Z}})$ | $\frac{1}{n}\sum_{i=1}^{n}\sqrt{2I(W;U_i|\widetilde{\mathbf{Z}})}$ |
| Individually CIMI (ICIMI) | (Zhou et al., 2022a) | $I(W;U_i|\widetilde{Z}_i)$ | $\frac{1}{n}\sum_{i=1}^{n}\sqrt{2I(W;U_i|\widetilde{Z}_i)}$ |
| Single ICIMI (SICIMI) | Ours | $I(W;U_i|\widetilde{Z}_i^0)$ | $\frac{1}{n}\sum_{i=1}^{n}\sqrt{2I(W;U_i|\widetilde{Z}_i^0)}$ |

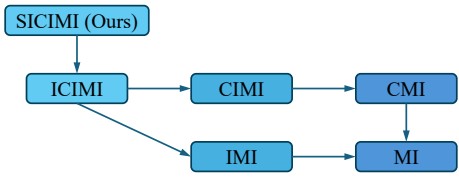

Figure 2: Comparison between approaches listed in Table 1. An arrow from A to B indicates that A is tighter than B.

**Theorem 3.2.** *Assume $\ell(\cdot,\cdot) \in [0,1]$, then*

$$d_{\mathrm{JS}}(L_n \,\|\, L_\mu) \leq \frac{1}{n}\sum_{i=1}^{n} I(W;U_i|\widetilde{Z}_i^0).$$

Theorem 3.2 exemplifies one of the major directions in information-theoretic generalization bounds, leveraging mutual information measures that involve the hypothesis $W$. This line of inquiry originated with the foundational work of (Xu & Raginsky, 2017), which introduced the idea of bounding generalization error through the mutual information $I(W;\mathbf{Z})$ between the hypothesis and the training dataset. Subsequent efforts have extended this framework, as seen in (Bu et al., 2020; Hellström & Durisi, 2022b), by incorporating individual or conditional information measures.

Table 1 provides a summary of recent hypothesis-based approaches to information-theoretic generalization analysis, focusing on the square-root bound for the specific case where $\ell(\cdot,\cdot) \in [0,1]$, chosen for simplicity. For equitable comparison, we demonstrate that our bound can be straightforwardly relaxed to yield a square-root bound:

**Corollary 3.3.** *Assume $\ell(\cdot,\cdot) \in [0,1]$, then*

$$|\overline{\mathrm{gen}}| \leq \frac{1}{n}\sum_{i=1}^{n}\sqrt{2I(W;U_i|\widetilde{Z}_i^0)}.$$

An intuitive comparison of the tightness of these bounds is provided in Figure 2. Examining the progression of works (CMI → CIMI → ICIMI) suggests that bounds may be

improved by eliminating redundant random variables from the key mutual information terms. Building on this insight, Corollary 3.3 reduces the number of conditional samples from 2 ($\widetilde{Z}_i$) to 1 ($\widetilde{Z}_i^0$) compared to the prior work of (Zhou et al., 2022a), thereby sharpening the bound. We designate this method as **Single Individually Conditional Individual Mutual Information (SICIMI)**, whose superiority is formally established in the following proposition:

**Proposition 3.4.** *For any $i \in [1,n]$, we have*

$$I(W;U_i|\widetilde{Z}_i^0) \leq I(W;U_i|\widetilde{Z}_i).$$

The above proposition demonstrates that our SICIMI bound in Theorem 3.2 is strictly tighter than all hypothesis-based information-theoretic generalization bounds summarized in Table 1. The following example illustrates that our SICIMI bound can be exactly tight for specific learning algorithms:

**Example 3.5.** Consider the case with $n = 1$, $\mu = \mathrm{Bern}\left(\frac{1}{2}\right)$, $W = \widetilde{Z}_1^{U_1}$ and $\ell(w,z) = \mathbb{1}_{w=z}$. One can verify that:

$$L_n = 0, \quad L_\mu = \tfrac{1}{2}, \quad \rightarrow \quad \overline{\mathrm{gen}} = \tfrac{1}{2}.$$
$$d_{\mathrm{JS}}(L_n \,\|\, L_\mu) = I(W;U_1|\widetilde{Z}_1^0) \approx 0.216.$$
$$I(W;U_1|\widetilde{Z}_1) = \tfrac{\log 2}{2} \approx 0.347 > d_{\mathrm{JS}}(L_n \,\|\, L_\mu).$$

Moreover, further eliminating the dependence on $\widetilde{Z}_i^0$ in these bounds will be infeasible, as it can be readily verified that $I(W;U_i) = 0$ in the example above. These observations collectively suggest that our SICIMI approach is optimal in this line of research, which tries to capture the behavior of the generalization error through mutual information measures involving only $W$, $U$, and $\widetilde{\mathbf{Z}}$.

### 3.2. A Prediction-based Generalization Bound

Now consider another case where $X = L_i^0$, $Y = U_i$, and $f(l) = l$. Building on Lemma 3.1, we established that:

**Theorem 3.6.** *Assume $\ell(\cdot,\cdot) \in [0,1]$, then*

$$d_{\mathrm{JS}}(L_n \,\|\, L_\mu) \leq \frac{1}{n}\sum_{i=1}^{n} I(L_i^0;U_i).$$

Theorem 3.6 represents another principal category of information-theoretic generalization bounds, leveraging mutual information measures involving network predictions. This direction can be traced back to (Harutyunyan et al., 2021), which introduced generalization bounds based on the mutual information $I(f_W(\tilde{Z}_i); U_i | \tilde{\mathbf{Z}})$ between supersample variables and network outputs (f-CMI). Subsequent refinements include evaluated losses (e-CMI) in (Steinke & Zakynthinou, 2020; Hellström & Durisi, 2022b) and loss differences (ld-CMI) or single-loss mutual information terms $I(L_i^0; U_i)$ in (Wang & Mao, 2023a).

In Figure 1, we compare our Binary JS bound (Theorem 3.6) with several related results in the literature, including the Square-Root bound (Theorem 4.1, (Wang & Mao, 2023a)), the Fast-Rate bound (Theorem 4.3, (Wang & Mao, 2023a)), and the Binary KL[1] bound (Theorem 5, (Hellström & Durisi, 2022b)). The population risk $L_\mu$ is evaluated within $[0, 1]$ using 0-1 loss to simplify mutual information calculations. As shown in Figure 1, our Binary JS bound provides the tightest estimation for the population risk. Although the figure highlights the case where $L_n = 0.2$, the superiority of Theorem 3.6 is consistent across any $L_n \in [0, 1]$. Furthermore, Theorem 3.6 consistently offers non-vacuous upper-bound estimations for any $L_\mu \in [L_n, 1]$, whereas other bounds yield vacuous results for $L_\mu \gtrsim 0.8$. Most notably, our Binary JS bound accurately recovers the true population risk when 0-1 loss is applied. This observation will be explored in detail in the following sections.

## 3.3. Exactly Tight Generalization Bounds

Next, we explore specific scenarios where the true generalization error can be precisely recovered using our Binary JS bound. To this end, we outline the assumptions used throughout this section:

**Assumption 3.7.** $L_n \leq L_\mu$.

We assume that the expected empirical risk is less than or equal to the expected population risk. This assumption is naturally satisfied by any well-trained network.

**Assumption 3.8.** $\mathcal{A}$ is invariant to sample permutations, i.e. $P_{W,U_i,\tilde{Z}_i} = P_{W,U_j,\tilde{Z}_j}$ for any $i, j \in [1, n]$.

We further assume that the learning algorithm $\mathcal{A}$ is invariant to the indices of samples. This property is typically satisfied by most noisy and iterative learning algorithms, such as SGD or Adam, since mini-batches are selected uniformly. It also holds true for full-batch gradient descent.

### 3.3.1. BINARY LOSS FUNCTION

We first show that for binary loss functions, such as the 0-1 loss, the Binary JS bound in Theorem 3.6 is exactly tight:

---

[1] For this evaluation, we approximate $I(L_i; U_i) \approx 2I(L_i^0; U_i)$.

1. Observe that the function $f(l) = l$ with $l \in \{0, 1\}$ satisfies the additional conditions of Lemma 3.1. Consequently, Eq. (3) holds with equality.
2. The joint convexity of $d_{\mathrm{KL}}(\cdot \| \cdot)$ directly implies the joint convexity of $d_{\mathrm{JS}}(\cdot \| \cdot)$. Since $d_{\mathrm{JS}}(p \| q)$ attains its minimum when $p = q$, Eq. (4) provides a unique solution for $L_\mu$ under Assumption 3.7, ensuring that Eq. (5) holds with equality.
3. Finally, Assumption 3.8 guarantees that $I(L_i^0; U_i)$ is identical for all $i \in [1, n]$.

These observations collectively imply the following result:

**Theorem 3.9.** *Assume* $\ell(\cdot, \cdot) \in \{0, 1\}$, *then*

$$L_\mu = d_{\mathrm{JS}}^{-1}\left(L_n, \frac{1}{n}\sum_{i=1}^n I(L_i^0; U_i)\right).$$

This result indicates that the single-loss mutual information term $I(L_i^0; U_i)$ is sufficient to characterize the true generalization error precisely. Importantly, it suggests that introducing additional assumptions, such as hypothesis stability (Wang & Mao, 2023b), is unnecessary for deriving vanishing information-theoretic generalization bounds.

### 3.3.2. BOUNDED LOSS FUNCTION

While the results above are promising, they do not generalize to the wide variety of loss functions commonly used in practice. To address this, we extend these results to general bounded continuous or discrete loss functions by employing a perturbation and rounding scheme for the evaluated loss variables. Let $D = \{(D_i^0, D_i^1)\}_{i=1}^n \sim \mathrm{Unif}([0, 1]^{2n})$ be independent random uniform variables. For any $i \in [1, n]$ and $u \in \{0, 1\}$, we define the binarized loss variables as:

$$\bar{L}_i^u = \lfloor L_i^u + D_i^u \rfloor, \quad \rightarrow \quad \bar{L}_i^u | L_i^u \sim \mathrm{Bern}(L_i^u).$$

When $\ell(\cdot, \cdot) \in [0, 1]$, it is straightforward to verify that $\bar{L}_i^u \in \{0, 1\}$ almost surely and satisfies $\mathbb{E}[\bar{L}_i^u] = \mathbb{E}[L_i^u]$. Thus, the empirical risk can be equivalently expressed as

$$L_n = \frac{1}{n}\sum_{i=1}^n \mathbb{E}[L_i^{U_i}] = \frac{1}{n}\sum_{i=1}^n \mathbb{E}[\bar{L}_i^{U_i}].$$

Similar results also apply to $L_\mu$. Consequently, it follows directly from Theorem 3.9 that:

**Corollary 3.10.** *Assume* $\ell(\cdot, \cdot) \in [0, 1]$, *then*

$$L_\mu = d_{\mathrm{JS}}^{-1}\left(L_n, \frac{1}{n}\sum_{i=1}^n I(\bar{L}_i^0; U_i)\right).$$

Given the Markov chain $U_i - L_i^0 - \bar{L}_i^0$, it follows from the data-processing inequality that $I(\bar{L}_i^0; U_i) \leq I(L_i^0; U_i)$. Thus, our **binarized-loss mutual information (bl-MI)** is

strictly tighter than the single-loss mutual information in Theorem 3.6 and precisely recovers the true generalization error for any bounded loss function.

Additionally, this perturbation and rounding technique is not confined to our binary JS bound. It can also be applied to any other prediction-based information-theoretic generalization measures, such as e-CMI (Hellström & Durisi, 2022b) or ld-MI (Wang & Mao, 2023a; 2024), to achieve tighter generalization bounds. Importantly, the bl-MI framework offers direct computational tractability, as $I(\bar{L}_i^0; U_i)$ only involves binary variables that can be directly estimated through binning. In contrast, estimating $I(L_i^0; U_i)$ for continuous $L_i^0$ is significantly more challenging, often requiring kernel density estimation techniques.

Recently, (Haghifam et al., 2023) demonstrated that most existing information-theoretic generalization bounds, including our Binary JS bound presented in Theorem 3.6, fail to vanish for certain stochastic convex optimization problems. Specifically, consider the following scenario:

**Example 3.11.** (Theorem 17, Haghifam et al. (2023)) Let $T = n^2$, $\eta = \frac{1}{n\sqrt{n}}$, and $d = 2n^2$. Define the instance space $\mathcal{Z} = \{e(i) : i \in d\}$, where $e(i)$ represents the $i$-th one-hot vector, and let $\mu = \text{Unif}(\mathcal{Z})$. A hypothesis is chosen from $\mathcal{W} = \{w \in \mathbb{R}^d : \|w\| \leq 1\}$ via full-batch gradient descent with the loss function $\ell(w, z) = 1 - \langle w, z \rangle$, using a learning rate $\eta$ over $T$ iterations.

When the samples in $\widetilde{\mathbf{Z}}$ are all distinct (which occurs with probability at least $\frac{1}{2}$), the empirical risk is $L_n = 1 - \frac{1}{\sqrt{n}}$, while the population risk is $L_\mu = 1 - \frac{1}{2n\sqrt{n}}$, yielding a generalization error $\overline{\text{gen}} = O(\frac{1}{\sqrt{n}})$. However, it can be verified that the supersample variables $U_i$ could be reconstructed by the information of $L_i^0$, leading to $I(L_i^0; U_i) \geq I(\mathbb{1}_{L_i^0 = 0}; U_i) = \log 2 = \Omega(1)$. This indicates that the generalization bound in Theorem 3.6 does not even vanish as $n \to \infty$, revealing that bounds relying on evaluated loss-based information measures may be loose for certain learning tasks, including e-CMI bounds (Hellström & Durisi, 2022b) and ld-CMI bounds (Wang & Mao, 2023a).

In contrast, by observing that $\ell(w, z) \in [0, 1]$ for any $\|w\|, \|z\| \leq 1$, our exactly tight bound in Corollary 3.10 remains valid in these scenarios, underscoring the advantages of our bl-MI measures. This improvement is achieved through our binarization technique, which eliminates extraneous information needed to reconstruct $U_i$ from the observations of $L_i^0$, thereby producing tighter information generalization measures. While recent works also provided vanishing bounds for Example 3.11 from the perspectives of hypothesis stability (Wang & Mao, 2023b) and algorithm compressibility (Sefidgaran & Zaidi, 2024), our results apply broadly to general learning algorithms without imposing any stability or compressibility requirements.

### 3.3.3. UNBOUNDED LOSS FUNCTION

Besides bounded loss functions, unbounded ones such as cross-entropy are also frequently encountered in practical scenarios. To address this, we extend the aforementioned results to accommodate general loss functions by employing a truncation-and-summing strategy. For any $i \in [1, n]$ and $u \in \{0, 1\}$, we define the truncated loss as:

$$\delta_j \bar{L}_i^u = \mathbb{1}_{\bar{L}_i^u \geq j}.$$

The corresponding truncated empirical risk is defined as $\delta_j L_n = \frac{1}{n} \sum_{i=1}^n \mathbb{E}[\delta_j \bar{L}_i^{U_i}]$, which can be verified to satisfy $L_n = \sum_{j=1}^\infty \delta_j L_n$. Similarly, the truncated population risk is denoted as $\delta_j L_\mu$. By recursively applying Corollary 3.10, the following result can be established:

**Corollary 3.12.** *Assume $\delta_j L_n \leq \delta_j L_\mu$ for all $j \geq 1$, then*

$$L_\mu = \sum_{j=1}^\infty d_{\text{JS}}^{-1}\left( \delta_j L_n, \frac{1}{n} \sum_{i=1}^n I(\delta_j \bar{L}_i^0; U_i) \right).$$

Throughout this section, we assume a loss interval of $[0, 1]$ for simplicity. However, these results can be straightforwardly extended to any interval by applying a linear transformation to both the loss variables and the bounds. Additionally, non-uniform truncation thresholds in Corollary 3.12 can be chosen to enhance flexibility, enabling tailored adjustments to specific requirements.

## 4. Extending beyond KL Divergence

It is well-established that the widely used KL divergence is a specific instance of the broader family of $f$-divergences, with $f(x) = x \log x$. In this section, we extend our previous results to accommodate arbitrary $f$-divergence measures. Given a convex function $f : [0, +\infty) \mapsto (-\infty, +\infty)$ which satisfies that $f(x)$ is finite for any $x > 0$ and $f(1) = 0$, the $f$-divergence between absolutely continuous probability distributions $P$ and $Q$ is defined as:

$$D_f(P \| Q) \triangleq \mathbb{E}_Q\left[f\left(\frac{dP}{dQ}\right)\right].$$

Following (Wang & Mao, 2024), the concept of mutual information can be generalized to $f$-information as:

$$I_f(X; Y) \triangleq D_f(P_{X,Y} \| P_X P_Y).$$

Analogous to Eq. (1) and (2), we define the $f$-divergence counterparts for binary KL and JS divergences as follows:

$$d_f(p \| q) \triangleq qf\left(\frac{p}{q}\right) + (1 - q)f\left(\frac{1-p}{1-q}\right).$$

$$d_{f\text{-JS}}(p \| q) \triangleq \frac{1}{2}d_f\left(p \| \frac{p+q}{2}\right) + \frac{1}{2}d_f\left(q \| \frac{p+q}{2}\right).$$

Using these definitions, we extend our main Lemma 3.1 to accommodate $f$-divergence measures:

**Lemma 4.1.** *Given random variables* $X$, $Y$ *such that* $Y \sim \mathrm{Bern}\left(\frac{1}{2}\right)$ *and* $X \in \{0, 1\}$ *almost surely, then*

$$d_{f\text{-JS}}\big(\mathbb{E}_{X|Y=0}[X] \,\big\|\, \mathbb{E}_{X|Y=1}[X]\big) = I_f(X; Y).$$

### 4.1. A Exactly Tight $f$-Information Bound

Building upon Lemma 4.1, we extend the previous exactly tight bound in Corollary 3.10 to incorporate $f$-information quantities. Specifically, under Assumption 3.8, we establish the following result:

**Theorem 4.2.** *Assume* $\ell(\cdot, \cdot) \in [0, 1]$, *then*

$$d_{f\text{-JS}}(L_n \,\|\, L_\mu) = \frac{1}{n} \sum_{i=1}^{n} I_f(\bar{L}_i^0; U_i).$$

It is noteworthy that the binary $f$-JS divergence in the above equation retains joint convexity, as $f$-divergence itself is jointly convex (Theorem 7.5, Polyanskiy & Wu (2024)). Consequently, under Assumption 3.7, the value of $L_\mu$ can be uniquely determined by $L_n$ and $d_{f\text{-JS}}(L_n \,\|\, L_\mu)$, in a manner analogous to Eq. (4).

### 4.2. A Exactly Tight Wasserstein Distance Bound

Inspired by (Rodríguez Gálvez et al., 2021), which demonstrates that generalization bounds based on Wasserstein distance metrics are generally tighter than those derived from information-theoretic measures, we further derive the following exactly tight bound using the Wasserstein distance between conditional and marginal loss distributions:

**Corollary 4.3.** *Assume* $\ell(\cdot, \cdot) \in [0, 1]$, *then*

$$|\overline{\mathrm{gen}}| = \frac{2}{n} \sum_{i=1}^{n} \mathbb{E}_{U_i}\left[\mathbb{W}\left(P_{\bar{L}_i^0|U_i}, P_{\bar{L}_i^0}\right)\right].$$

The result above is a direct application of Theorem 4.2 to the case of total variation, noting that the Wasserstein distance $\mathbb{W}$ and total variation distance $D_{\mathrm{TV}}$ are equivalent for discrete distributions $P$ and $Q$:

$$D_{\mathrm{TV}}(P \,\|\, Q) = \mathbb{W}(P, Q).$$

## 5. Related Works and Limitations

In addition to the works of information-theoretic generalization bounds discussed earlier, significant progress has been made in enhancing these bounds through conditioning (Hafez-Kolahi et al., 2020) and chaining techniques (Asadi et al., 2018; Zhou et al., 2022b; Clerico et al., 2022). A noteworthy development within the supersample framework is the leave-one-out setting (Haghifam et al., 2022; Rammal et al., 2022), which reduces the sample size requirement

from $2n$ to $n + 1$. Beyond traditional supervised learning, information-theoretic bounds have been extended to analyze generalization across meta-learning (Rezazadeh et al., 2021; Jose et al., 2021; Hellström & Durisi, 2022a), semi-supervised learning (Aminian et al., 2022; He et al., 2022), and transfer learning (Wu et al., 2020; Masiha et al., 2021; Wang & Mao, 2022; Bu et al., 2022) paradigms. Furthermore, recent works (Haghifam et al., 2023; Livni, 2024; Attias et al., 2024) have highlighted the limitations of existing information-theoretic bounds in certain stochastic convex optimization settings. This paper adequately addresses these issues through the proposed ld-MI measure, which resolves the identified failures and provides an exactly tight bound for general learning algorithms.

While we use the binary JS divergence as the comparator between empirical and population losses, (Hellström & Guedj, 2024) explores a general convex comparator for this purpose and further investigates the optimal convex comparator. According to their analysis, the optimal comparator for the standard generalization analysis setting is the binary KL divergence when $\ell(\cdot, \cdot) \in [0, 1]$. However, their conclusion does not naturally extends to the supersample framework, as we have clearly shown that our binary JS outperforms binary KL. This stems from the fact that the key information quantities for supersample settings are $I(L_i, U_i)$, while those for the standard setting are $I(W; \mathbf{Z})$. This may explain, from another perspective, why our binary JS method is limited to the supersample setting.

The limitations of this work are as follows: While Corollary 3.12 provides an exact characterization of the generalization error for general unbounded loss functions, the precondition $\delta_j L_n \leq \delta_j L_\mu$ for each $j \geq 1$ is significantly more stringent than Assumption 3.7 and may not always be satisfied in practice. Additionally, the primary focus of this paper is on expected generalization bounds. It remains a compelling avenue for future research to develop exactly tight bounds that hold with high probability, addressing a broader spectrum of practical scenarios.

## 6. Experimental Results

In this section, we assess the tightness of our exactly tight Binary JS bound (Corollary 3.10) in comparison to several existing information-theoretic generalization bounds from the literature. These include the Fast-Rate bound (Theorem 4.3, (Wang & Mao, 2023a)), the Binary KL bound (Theorem 5, (Hellström & Durisi, 2022b)), and the $f$-information series of oracle bounds: CMI, CSHI, and CJSI (Theorems 3.1, 3.2, and 3.3, (Wang & Mao, 2024)). Our experimental settings align closely with those in (Wang & Mao, 2024), where we evaluate three distinct classification tasks[2]:

[2] https://github.com/Yuxin-Dong/BinaryJS.

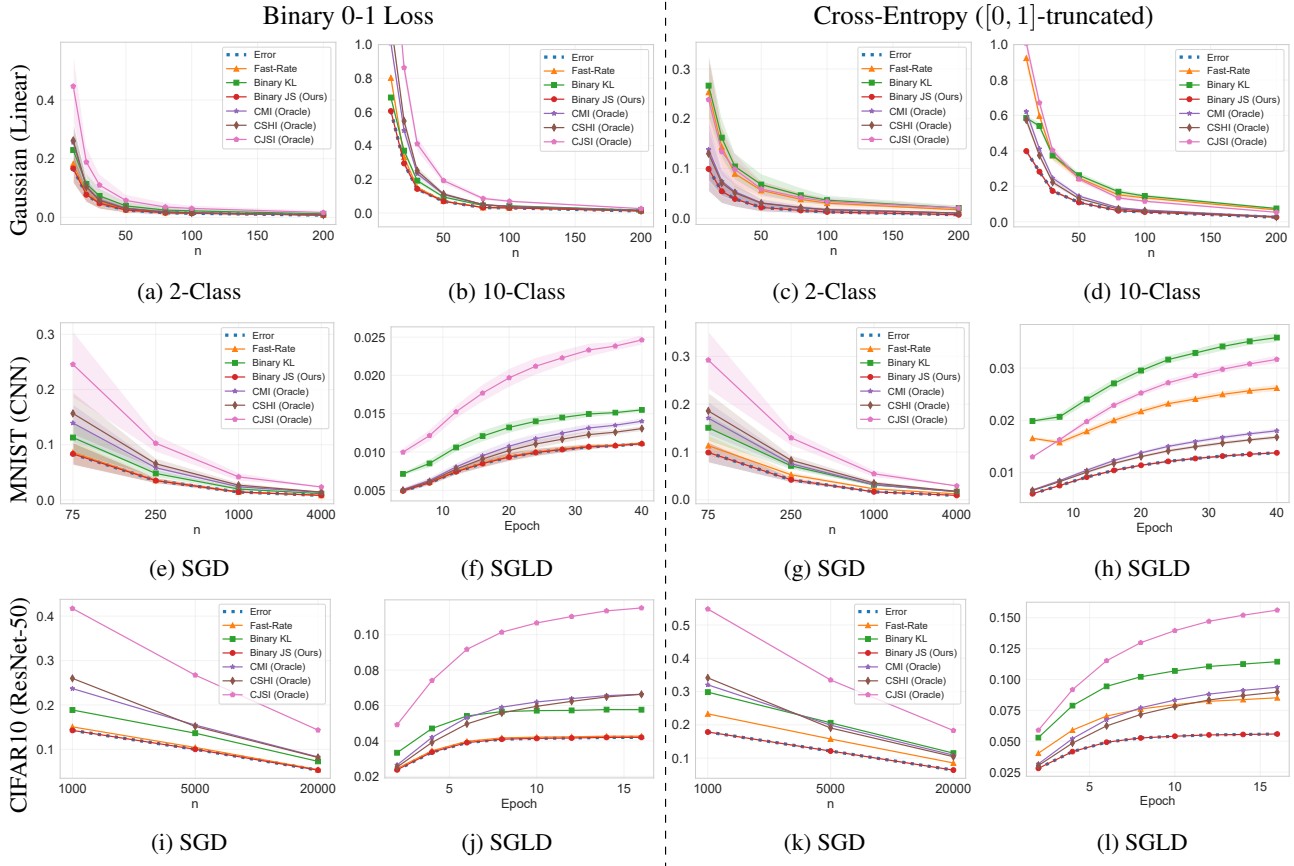

Figure 3: Comparison of generalization bounds under different learning scenarios. (a-d) Linear classifier trained on synthetic Gaussian dataset. (e-h) CNN trained on binary MNIST (4 vs. 9). (i-l) Pretrained ResNet-50 fine-tuned on CIFAR10.

- Simple linear classifier on synthetic Gaussian dataset.
- 4-layer CNN on binarized MNIST (classes 4 vs. 9).
- Pretrained ResNet-50 model on CIFAR10.

In all these scenarios, Assumption 3.8 is inherently satisfied, since the linear classifier is trained with full-batch gradient descent while CNN and ResNet-50 models are trained with mini-batch-based iterative learning algorithms such as SGD and SGLD. Moreover, Assumption 3.7 is empirically validated in every experiment conducted.

To evaluate the effectiveness of generalization bounds for both discrete and continuous loss variables, we consider two distinct loss functions: 1) The binary 0-1 loss. 2) A truncated version of the cross-entropy loss constrained within $[0, 1]$. This approach reflects practical scenarios, as cross-entropy loss is typically clamped to a constant range to ensure numerical stability. For the binary 0-1 loss, we employ a simple binning method to estimate the probability distribution of $P_{L_i^0, U_i}$. For the truncated cross-entropy loss, we use probability density estimation via truncated Gaussian kernels, with bandwidth determined by the rule-of-thumb criterion. To satisfy Assumption 3.8, we jointly utilize the

samples of every $(L_i^0, U_i)$, $i \in [1, n]$ to estimate the unified mutual information $I(L_i^0; U_i)$. This method deviates slightly from prior approaches (Dong et al., 2024b;a; Wang & Mao, 2024). However, we argue that this modification does not introduce additional bias to the estimated probability distribution, as discussed in Appendix C.

The final results, presented in Figure 3, demonstrate that our Binary JS bound fully captures the dynamics of the generalization error. When using the 0-1 loss, the Fast-Rate bound also closely tracks the generalization gap, differing only slightly from our Binary JS bound. This similarity arises because both bounds leverage the single-loss mutual information $I(L_i^0; U_i)$. As shown in Figure 1, the difference between these two is negligible when $I(L_i^0; U_i)$ is small. However, for continuous loss functions, the gap becomes pronounced, underscoring the superiority of our bl-MI approach. In the case of the SGLD algorithm, the CMI and CSHI bounds effectively utilize the scale information of the loss variables to produce tighter estimates. Lastly, across all the practical learning scenarios considered, our Binary JS bound consistently achieves exact tightness relative to the true generalization error.

# 7. Conclusion

In this paper, we introduce a new framework for characterizing the relationship between expected empirical and population risks via the binarized Jensen-Shannon divergence. Our proposed bounds achieve exact tightness relative to the true generalization error for binary loss functions. Leveraging a binarization technique for supersample loss variables, we derive exactly tight generalization bounds applicable to general learning algorithms under mild assumptions. These results effectively resolve the limitations of prior information-theoretic bounds, particularly their failure in certain stochastic convex optimization scenarios.

# Acknowledgements

This work was supported by the National Natural Science Foundation of China (62172326) and the Fundamental Research Funds for the Central Universities (xxj032025002). We would like to thank the anonymous reviewers for their valuable suggestions and for bringing the related work of (Hellström & Guedj, 2024) to our attention.

# Impact Statement

This paper presents work whose goal is to advance the field of Machine Learning. There are many potential societal consequences of our work, none of which we feel must be specifically highlighted here.

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

## A. Prerequisite Definitions and Lemmas

**Definition A.1.** (Sub-Gaussian) A random variable $X$ is $\sigma$-sub-Gaussian if for any $\rho \in \mathbb{R}$, $\mathbb{E}[e^{\rho(X-\mathbb{E}[X])}] \leq \exp(\frac{\rho^2 \sigma^2}{2})$.

**Definition A.2.** (Kullback-Leibler Divergence) Let $P$ and $Q$ be probability measures on the same space $\mathcal{X}$, then the KL divergence from $P$ to $Q$ is defined as $D_{\mathrm{KL}}(P \parallel Q) \triangleq \int_{\mathcal{X}} p(x) \log\left(\frac{p(x)}{q(x)}\right) \mathrm{d}x$.

**Definition A.3.** (Mutual Information) Let $(X, Y)$ be a pair of random variables with values over the space $\mathcal{X} \times \mathcal{Y}$. Let their joint distribution be $P_{X,Y}$ and the marginal distributions be $P_X$ and $P_Y$ respectively, then the mutual information between $X$ and $Y$ is defined as $I(X; Y) = D_{\mathrm{KL}}(P_{X,Y} \parallel P_X P_Y)$.

**Definition A.4.** (Wasserstein Distance) Let $c(\cdot, \cdot)$ be a metric and let $P$ and $Q$ be probability measures on $\mathcal{X}$. Denote $\Gamma(P, Q)$ as the set of all couplings of $P$ and $Q$ (i.e. the set of all joint distributions on $\mathcal{X} \times \mathcal{X}$ with two marginals being $P$ and $Q$), then the Wasserstein distance of order $p$ between $P$ and $Q$ is defined as $\mathbb{W}_p(P, Q) \triangleq \left(\inf_{\gamma \in \Gamma(P,Q)} \int_{\mathcal{X} \times \mathcal{X}} c(x, x')^p \, \mathrm{d}\gamma(x, x')\right)^{1/p}$.

Unless otherwise noted, we use $\mathbb{W}(\cdot, \cdot)$ to denote the Wasserstein distance of order $1$.

**Lemma A.5.** *(Donsker-Varadhan formula) Let $P$ and $Q$ be probability measures defined on the same measurable space $\mathcal{X}$, where $P$ is absolutely continuous with respect to $Q$. Then for any bounded measurable function $f : \mathcal{X} \mapsto \mathbb{R}$,*

$$D(P \parallel Q) = \sup_f \left\{ \mathbb{E}_{x \sim P}[f(x)] - \log \mathbb{E}_{x \sim Q}[e^{f(x)}] \right\},$$

*where $X$ is any random variable such that $e^X$ is $Q$-integrable and $\mathbb{E}_P[X]$ exists.*

**Lemma A.6.** *(Lemma 2, Hellström & Durisi (2022b)) Let $X$ be a random variable that satisfies $X \in [0, 1]$ almost surely and $\mathbb{E}[X] = \mu$. Then for any $\gamma \in \mathbb{R}$,*

$$\mathbb{E}_X \left[ e^{d_\gamma(X \parallel \mu)} \right] \leq 1.$$

**Lemma A.7.** *(Pinsker's Inequality) Let $P$ and $Q$ be probability measures defined on the same space, then $D_{\mathrm{TV}}(P \parallel Q) \leq \sqrt{\frac{1}{2} D_{\mathrm{KL}}(P \parallel Q)}$.*

## B. Omitted Proofs

**Lemma 3.1** (Restate). *Given random variable $X$, binary variable $Y \sim \mathrm{Bern}\left(\frac{1}{2}\right)$ and a measurable function $f(x)$. Assume that $f(X) \in [0, 1]$ almost surely, then*

$$d_{\mathrm{JS}}\big(\mathbb{E}_{X|Y=0}[f(X)] \,\big\|\, \mathbb{E}_{X|Y=1}[f(X)]\big) \leq I(X; Y).$$

*Additionally, if $f(x)$ is invertible and $f(X) \in \{0, 1\}$ almost surely, then the inequality above holds with equality.*

*Proof.* By Jensen's inequality and the joint convexity of $d_\gamma(\cdot \parallel \cdot)$, we have

$$
\begin{aligned}
d_{\mathrm{KL}}\big(\mathbb{E}_{X|Y=0}[f(X)] \,\big\|\, \mathbb{E}_X[f(X)]\big) &= \sup_\gamma d_\gamma\big(\mathbb{E}_{X|Y=0}[f(X)] \,\big\|\, \mathbb{E}_X[f(X)]\big) \\
&\leq \sup_\gamma \mathbb{E}_{X|Y=0}[d_\gamma(f(X) \,\|\, \mathbb{E}_X[f(X)])].
\end{aligned}
\tag{6}
$$

Similarly, we can prove that

$$d_{\mathrm{KL}}\big(\mathbb{E}_{X|Y=1}[f(X)] \,\big\|\, \mathbb{E}_X[f(X)]\big) \leq \sup_\gamma \mathbb{E}_{X|Y=1}[d_\gamma(f(X) \,\|\, \mathbb{E}_X[f(X)])].$$

Notice that $\mathbb{E}_X[f(X)] = \frac{\mathbb{E}_{X|Y=0}[f(X)] + \mathbb{E}_{X|Y=1}[f(X)]}{2}$, we then have

$$
\begin{aligned}
d_{\mathrm{JS}}\big(\mathbb{E}_{X|Y=0}[f(X)] \,\big\|\, \mathbb{E}_{X|Y=1}[f(X)]\big) &= \frac{1}{2} d_{\mathrm{KL}}\big(\mathbb{E}_{X|Y=0}[f(X)] \,\big\|\, \mathbb{E}_X[f(X)]\big) + \frac{1}{2} d_{\mathrm{KL}}\big(\mathbb{E}_{X|Y=1}[f(X)] \,\big\|\, \mathbb{E}_X[f(X)]\big) \\
&\leq \frac{1}{2} \sum_{u \in \{0,1\}} \sup_\gamma \mathbb{E}_{X|Y=u}[d_\gamma(f(X) \,\|\, \mathbb{E}_X[f(X)])].
\end{aligned}
\tag{7}
$$

By applying Lemma A.5 with $P = P_{X|Y=0}$, $Q = P_X$ and $f(x, y) = d_\gamma(f(x) \| \mathbb{E}_X[f(X)])$, we have

$$D_{\mathrm{KL}}\big(P_{X|Y=0} \,\big\|\, P_X\big) \geq \mathbb{E}_{X|Y=0}[d_\gamma(f(X) \| \mathbb{E}_X[f(X)])] - \log \mathbb{E}_X\Big[e^{d_\gamma(f(X) \| \mathbb{E}_X[f(X)])}\Big].$$

Then by applying Lemma A.6 we know that for any $\gamma \in \mathbb{R}$,

$$\mathbb{E}_X\Big[e^{d_\gamma(f(X) \| \mathbb{E}_X[f(X)])}\Big] \leq 1.$$

Plugging this into the inequality above and taking the supremum over $\gamma$, we can get

$$D_{\mathrm{KL}}\big(P_{X|Y=0} \,\big\|\, P_X\big) \geq \sup_\gamma \mathbb{E}_{X|Y=0}[d_\gamma(f(X) \| \mathbb{E}_X[f(X)])]. \tag{8}$$

Similarly, we have the same result for the case $Y = 1$ as

$$D_{\mathrm{KL}}\big(P_{X|Y=1} \,\big\|\, P_X\big) \geq \sup_\gamma \mathbb{E}_{X|Y=1}[d_\gamma(f(X) \| \mathbb{E}_X[f(X)])].$$

Combining the two inequalities above, we now have

$$
\begin{aligned}
I(X;Y) &= \mathbb{E}_Y\big[D_{\mathrm{KL}}\big(P_{X|Y} \,\big\|\, P_X\big)\big] \\
&= \frac{1}{2} D_{\mathrm{KL}}\big(P_{X|Y=0} \,\big\|\, P_X\big) + \frac{1}{2} D_{\mathrm{KL}}\big(P_{X|Y=1} \,\big\|\, P_X\big) \\
&\geq \frac{1}{2} \sum_{u \in \{0,1\}} \sup_\gamma \mathbb{E}_{X|Y=u}[d_\gamma(f(X) \| \mathbb{E}_X[f(X)])].
\end{aligned}
$$

Plugging the inequality above into Eq. (7), we finally get

$$d_{\mathrm{JS}}\big(\mathbb{E}_{X|Y=0}[f(X)] \,\big\|\, \mathbb{E}_{X|Y=1}[f(X)]\big) \leq \frac{1}{2} \sum_{u \in \{0,1\}} \sup_\gamma \mathbb{E}_{X|Y=u}[d_\gamma(f(X) \| \mathbb{E}_X[f(X)])] \leq I(X;Y).$$

The proof of the first part is complete.

To prove that the main inequality of the lemma holds with equality, we only need to check if the inequalities in Eq. (6) and (8) hold with equality. When $X \in \{0, 1\}$, we have $\mathbb{E}_{X|Y=0}[X] = P(X = 1|Y = 0)$ and

$$
\begin{aligned}
d_{\mathrm{KL}}\big(\mathbb{E}_{X|Y=0}[X] \,\big\|\, \mathbb{E}_X[X]\big) &= \sup_\gamma d_\gamma\big(\mathbb{E}_{X|Y=0}[X] \,\big\|\, \mathbb{E}_X[X]\big) \\
&= \sup_\gamma P(X = 1|Y = 0)\gamma - \log(1 - \mathbb{E}_X[X] + \mathbb{E}_X[X]e^\gamma) \\
&= \sup_\gamma P(X = 0|Y = 0)(0\gamma - \log(1 - \mathbb{E}_X[X] + \mathbb{E}_X[X]e^\gamma)) \\
&\quad + P(X = 1|Y = 0)(1\gamma - \log(1 - \mathbb{E}_X[X] + \mathbb{E}_X[X]e^\gamma)) \\
&= \sup_\gamma \mathbb{E}_{X|Y=0}\big[d_\gamma\big(\mathbb{E}_{X|Y=0}[X] \,\big\|\, \mathbb{E}_X[X]\big)\big].
\end{aligned}
$$

On the other hand, we can prove that

$$
\begin{aligned}
D_{\mathrm{KL}}\big(P_{X|Y=0} \,\big\|\, P_X\big) &= d_{\mathrm{KL}}(P(X = 0|Y = 0) \,\|\, P(X = 0)) \\
&= d_{\mathrm{KL}}(P(X = 1|Y = 0) \,\|\, P(X = 1)) \\
&= d_{\mathrm{KL}}\big(\mathbb{E}_{X|Y=0}[X] \,\big\|\, \mathbb{E}_X[X]\big).
\end{aligned}
$$

The same results also hold for $Y = 1$. Combining the results above, we then have

$$d_{\mathrm{JS}}\big(\mathbb{E}_{X|Y=0}[X] \,\big\|\, \mathbb{E}_{X|Y=1}[X]\big) = \frac{1}{2} \sum_{u \in \{0,1\}} D_{\mathrm{KL}}\big(P_{X|Y=u} \,\big\|\, P_X\big) = I(X;Y).$$

Therefore, when $f(X) \in \{0, 1\}$ almost surely, we have

$$d_{\mathrm{JS}}\big(\mathbb{E}_{X|Y=0}[f(X)] \,\big\|\, \mathbb{E}_{X|Y=1}[f(X)]\big) = I(f(X);Y).$$

Combining with the condition that $f(X)$ is invertible, we have $I(f(X);Y) = I(X;Y)$ by the data-processing inequality, which finishes the proof. $\square$

**Theorem 3.2** (Restate). *Assume $\ell(\cdot, \cdot) \in [0, 1]$, then*

$$d_{\mathrm{JS}}(L_n \,\|\, L_\mu) \leq \frac{1}{n} \sum_{i=1}^{n} I(W; U_i | \widetilde{Z}_i^0).$$

*Proof.* By Jensen's inequality and the joint convexity of $d_{\mathrm{KL}}(\cdot \,\|\, \cdot)$, we have

$$d_{\mathrm{KL}}\left(L_n \,\middle\|\, \frac{L_n + L_\mu}{2}\right) = d_{\mathrm{KL}}\left(\frac{1}{n}\sum_{i=1}^{n} \mathbb{E}_{W,\widetilde{Z}_i,U_i}\left[\ell(W, \widetilde{Z}_i^{U_i})\right] \,\middle\|\, \frac{1}{2n}\sum_{i=1}^{n} \mathbb{E}_{W,\widetilde{Z}_i}\left[\ell(W, \widetilde{Z}_i^0) + \ell(W, \widetilde{Z}_i^1)\right]\right)$$

$$\leq \frac{1}{n}\sum_{i=1}^{n} d_{\mathrm{KL}}\left(\mathbb{E}_{W,\widetilde{Z}_i,U_i}\left[\ell(W, \widetilde{Z}_i^{U_i})\right] \,\middle\|\, \frac{1}{2}\mathbb{E}_{W,\widetilde{Z}_i}\left[\ell(W, \widetilde{Z}_i^0) + \ell(W, \widetilde{Z}_i^1)\right]\right)$$

$$= \frac{1}{n}\sum_{i=1}^{n} d_{\mathrm{KL}}\left(\frac{1}{2}\mathbb{E}_{W,\widetilde{Z}_i^0 | U_i = 0}\left[\ell(W, \widetilde{Z}_i^0)\right] + \frac{1}{2}\mathbb{E}_{W,\widetilde{Z}_i^1 | U_i = 1}\left[\ell(W, \widetilde{Z}_i^1)\right]\right.$$

$$\left.\,\middle\|\, \frac{1}{2}\mathbb{E}_{W,\widetilde{Z}_i^0}\left[\ell(W, \widetilde{Z}_i^0)\right] + \frac{1}{2}\mathbb{E}_{W,\widetilde{Z}_i^1}\left[\ell(W, \widetilde{Z}_i^1)\right]\right)$$

$$\leq \frac{1}{2n}\sum_{i=1}^{n} d_{\mathrm{KL}}\left(\mathbb{E}_{W,\widetilde{Z}_i^0 | U_i = 0}\left[\ell(W, \widetilde{Z}_i^0)\right] \,\middle\|\, \mathbb{E}_{W,\widetilde{Z}_i^0}\left[\ell(W, \widetilde{Z}_i^0)\right]\right)$$

$$+ \frac{1}{2n}\sum_{i=1}^{n} d_{\mathrm{KL}}\left(\mathbb{E}_{W,\widetilde{Z}_i^1 | U_i = 1}\left[\ell(W, \widetilde{Z}_i^1)\right] \,\middle\|\, \mathbb{E}_{W,\widetilde{Z}_i^1}\left[\ell(W, \widetilde{Z}_i^1)\right]\right)$$

$$\leq \frac{1}{2n}\sum_{i=1}^{n} \mathbb{E}_{\widetilde{Z}_i^0}\left[d_{\mathrm{KL}}\left(\mathbb{E}_{W | \widetilde{Z}_i^0, U_i = 0}\left[\ell(W, \widetilde{Z}_i^0)\right] \,\middle\|\, \mathbb{E}_{W | \widetilde{Z}_i^0}\left[\ell(W, \widetilde{Z}_i^0)\right]\right)\right]$$

$$+ \frac{1}{2n}\sum_{i=1}^{n} \mathbb{E}_{\widetilde{Z}_i^1}\left[d_{\mathrm{KL}}\left(\mathbb{E}_{W | \widetilde{Z}_i^1, U_i = 1}\left[\ell(W, \widetilde{Z}_i^1)\right] \,\middle\|\, \mathbb{E}_{W | \widetilde{Z}_i^1}\left[\ell(W, \widetilde{Z}_i^1)\right]\right)\right].$$

Similarly, one can prove that

$$d_{\mathrm{KL}}\left(L_\mu \,\middle\|\, \frac{L_n + L_\mu}{2}\right) \leq \frac{1}{2n}\sum_{i=1}^{n} \mathbb{E}_{\widetilde{Z}_i^0}\left[d_{\mathrm{KL}}\left(\mathbb{E}_{W | \widetilde{Z}_i^0, U_i = 1}\left[\ell(W, \widetilde{Z}_i^0)\right] \,\middle\|\, \mathbb{E}_{W | \widetilde{Z}_i^0}\left[\ell(W, \widetilde{Z}_i^0)\right]\right)\right]$$

$$+ \frac{1}{2n}\sum_{i=1}^{n} \mathbb{E}_{\widetilde{Z}_i^1}\left[d_{\mathrm{KL}}\left(\mathbb{E}_{W | \widetilde{Z}_i^1, U_i = 0}\left[\ell(W, \widetilde{Z}_i^1)\right] \,\middle\|\, \mathbb{E}_{W | \widetilde{Z}_i^1}\left[\ell(W, \widetilde{Z}_i^1)\right]\right)\right].$$

Notice that

$$\mathbb{E}_{W | \widetilde{Z}_i^0}\left[\ell(W, \widetilde{Z}_i^0)\right] = \frac{1}{2}\left(\mathbb{E}_{W | \widetilde{Z}_i^0, U_i = 0}\left[\ell(W, \widetilde{Z}_i^0)\right] + \mathbb{E}_{W | \widetilde{Z}_i^0, U_i = 1}\left[\ell(W, \widetilde{Z}_i^0)\right]\right),$$

$$\mathbb{E}_{W | \widetilde{Z}_i^1}\left[\ell(W, \widetilde{Z}_i^1)\right] = \frac{1}{2}\left(\mathbb{E}_{W | \widetilde{Z}_i^1, U_i = 0}\left[\ell(W, \widetilde{Z}_i^1)\right] + \mathbb{E}_{W | \widetilde{Z}_i^1, U_i = 1}\left[\ell(W, \widetilde{Z}_i^1)\right]\right).$$

Combining the results above and applying Lemma 3.1 with $X = W | \widetilde{Z}_i^0$ (or $X = W | \widetilde{Z}_i^1$), $Y = U_i$ and $f(W) = \ell(W, \widetilde{Z}_i^0)$ (or $f(W) = \ell(W, \widetilde{Z}_i^1)$), we can get

$$d_{\mathrm{JS}}(L_n \,\|\, L_\mu) = \frac{1}{2}d_{\mathrm{KL}}\left(L_n \,\middle\|\, \frac{L_n + L_\mu}{2}\right) + \frac{1}{2}d_{\mathrm{KL}}\left(L_\mu \,\middle\|\, \frac{L_n + L_\mu}{2}\right)$$

$$\leq \frac{1}{2n}\sum_{i=1}^{n} \mathbb{E}_{\widetilde{Z}_i^0}\left[d_{\mathrm{JS}}\left(\mathbb{E}_{W | \widetilde{Z}_i^0, U_i = 0}\left[\ell(W, \widetilde{Z}_i^0)\right] \,\middle\|\, \mathbb{E}_{W | \widetilde{Z}_i^0, U_i = 1}\left[\ell(W, \widetilde{Z}_i^0)\right]\right)\right]$$

$$+ \frac{1}{2n}\sum_{i=1}^{n} \mathbb{E}_{\widetilde{Z}_i^1}\left[d_{\mathrm{JS}}\left(\mathbb{E}_{W | \widetilde{Z}_i^1, U_i = 0}\left[\ell(W, \widetilde{Z}_i^1)\right] \,\middle\|\, \mathbb{E}_{W | \widetilde{Z}_i^1, U_i = 1}\left[\ell(W, \widetilde{Z}_i^1)\right]\right)\right] \tag{9}$$

$$\leq \frac{1}{2n} \sum_{i=1}^{n} \mathbb{E}_{\widetilde{Z}_i^0} \left[ I^{\widetilde{Z}_i^0}(W; U_i) \right] + \frac{1}{2n} \sum_{i=1}^{n} \mathbb{E}_{\widetilde{Z}_i^1} \left[ I^{\widetilde{Z}_i^1}(W; U_i) \right]$$

$$= \frac{1}{2n} \sum_{i=1}^{n} I(W; U_i | \widetilde{Z}_i^0) + I(W; U_i | \widetilde{Z}_i^1).$$

Following the formulation of the supersample setting, we know that the learning algorithm $\mathcal{A}$ is unaware of the supersample variables $U_i$. Therefore, it should be invariant w.r.t the supersamples $\widetilde{Z}_i$, i.e. the distributions $P_{W,U_i,\widetilde{Z}_i^0}$ and $P_{W,U_i,\widetilde{Z}_i^1}$ should satisfy certain symmetry such that

$$P_{W,\widetilde{Z}_i^0|U_i=0} = P_{W,\widetilde{Z}_i^1|U_i=1}, \quad P_{W,\widetilde{Z}_i^0|U_i=1} = P_{W,\widetilde{Z}_i^1|U_i=0}. \tag{10}$$

Therefore, we can conclude that $I(W; U_i | \widetilde{Z}_i^0) = I(W; U_i | \widetilde{Z}_i^1)$ and the proof is complete. $\qquad \square$

**Corollary 3.3** (Restate). *Assume $\ell(\cdot, \cdot) \in [0, 1]$, then*

$$|\overline{\mathrm{gen}}| \leq \frac{1}{n} \sum_{i=1}^{n} \sqrt{2I(W; U_i | \widetilde{Z}_i^0)}.$$

*Proof.* Applying Lemma A.7, we have that for any $p, q \in [0, 1]$,

$$d_{\mathrm{JS}}(p \parallel q) = \frac{1}{2} d_{\mathrm{KL}} \left( p \parallel \frac{p+q}{2} \right) + \frac{1}{2} d_{\mathrm{KL}} \left( q \parallel \frac{p+q}{2} \right)$$

$$\geq d_{\mathrm{TV}}^2 \left( p \parallel \frac{p+q}{2} \right) + d_{\mathrm{TV}}^2 \left( q \parallel \frac{p+q}{2} \right) = \frac{1}{2}(p-q)^2.$$

With the same technique to bound Eq. (9), we then have

$$\left| \mathbb{E}_{W|\widetilde{Z}_i^0,U_i=0} \left[ \ell(W, \widetilde{Z}_i^0) \right] - \mathbb{E}_{W|\widetilde{Z}_i^0,U_i=1} \left[ \ell(W, \widetilde{Z}_i^0) \right] \right| \leq \sqrt{2 d_{\mathrm{JS}} \left( \mathbb{E}_{W|\widetilde{Z}_i^0,U_i=0} \left[ \ell(W, \widetilde{Z}_i^0) \right] \parallel \mathbb{E}_{W|\widetilde{Z}_i^0,U_i=1} \left[ \ell(W, \widetilde{Z}_i^0) \right] \right)}$$

$$\leq \sqrt{2 I^{\widetilde{Z}_i^0}(W; U_i)}.$$

Similarly, we can prove

$$\left| \mathbb{E}_{W|\widetilde{Z}_i^1,U_i=1} \left[ \ell(W, \widetilde{Z}_i^1) \right] - \mathbb{E}_{W|\widetilde{Z}_i^1,U_i=0} \left[ \ell(W, \widetilde{Z}_i^1) \right] \right| \leq \sqrt{2 I^{\widetilde{Z}_i^1}(W; U_i)}.$$

By the definition of generalization error and Jensen's inequality, we obtain

$$|\overline{\mathrm{gen}}| = \left| \mathbb{E}_{W,\widetilde{\mathbf{Z}},U} \left[ L_{\widetilde{\mathbf{Z}}_U}(W) - L_{\widetilde{\mathbf{Z}}_{\overline{U}}}(W) \right] \right| = \frac{1}{n} \left| \mathbb{E}_{W,\widetilde{\mathbf{Z}},U} \left[ \sum_{i=1}^{n} \ell(W; \widetilde{Z}_i^{U_i}) - \ell(W; \widetilde{Z}_i^{\overline{U}_i}) \right] \right|$$

$$\leq \frac{1}{n} \sum_{i=1}^{n} \left| \mathbb{E}_{\widetilde{Z}_i,U} \left[ \mathbb{E}_{W|\widetilde{Z}_i,U} \left[ \ell(W, \widetilde{Z}_i^{U_i}) \right] - \mathbb{E}_{W|\widetilde{Z}_i,U_i} \left[ \ell(W, \widetilde{Z}_i^{\overline{U}_i}) \right] \right] \right|$$

$$\leq \frac{1}{2n} \sum_{i=1}^{n} \left| \mathbb{E}_{\widetilde{Z}_i^0} \left[ \mathbb{E}_{W|\widetilde{Z}_i^0,U_i=0} \left[ \ell(W, \widetilde{Z}_i^0) \right] - \mathbb{E}_{W|\widetilde{Z}_i^0,U_i=1} \left[ \ell(W, \widetilde{Z}_i^0) \right] \right] \right|$$

$$+ \frac{1}{2n} \sum_{i=1}^{n} \left| \mathbb{E}_{\widetilde{Z}_i^1} \left[ \mathbb{E}_{W|\widetilde{Z}_i^1,U_i=1} \left[ \ell(W, \widetilde{Z}_i^1) \right] - \mathbb{E}_{W|\widetilde{Z}_i^1,U_i=0} \left[ \ell(W, \widetilde{Z}_i^1) \right] \right] \right|$$

$$\leq \frac{1}{2n} \sum_{i=1}^{n} \mathbb{E}_{\widetilde{Z}_i^0} \left[ \sqrt{2 I^{\widetilde{Z}_i^0}(W; U_i)} \right] + \mathbb{E}_{\widetilde{Z}_i^1} \left[ \sqrt{2 I^{\widetilde{Z}_i^1}(W; U_i)} \right]$$

$$\leq \frac{1}{2n} \sum_{i=1}^{n} \sqrt{2 I(W; U_i | \widetilde{Z}_i^0)} + \sqrt{2 I(W; U_i | \widetilde{Z}_i^1)}$$

$$= \frac{1}{n} \sum_{i=1}^{n} \sqrt{2I(W;U_i|\widetilde{Z}_i^0)}.$$

The proof is complete. $\qquad\square$

**Proposition 3.4** (Restate). *For any $i \in [1, n]$, we have*

$$I(W;U_i|\widetilde{Z}_i^0) \leq I(W;U_i|\widetilde{Z}_i).$$

*Proof.* Since $\widetilde{Z}_i^0$, $\widetilde{Z}_i^1$ and $U_i$ are independent, we have $I(\widetilde{Z}_i^1;U_i|\widetilde{Z}_i^0) = 0$. Then by the chain rule of conditional mutual information, we can prove that

$$I(W;U_i|\widetilde{Z}_i^0) \leq I(W;U_i|\widetilde{Z}_i^0) + I(\widetilde{Z}_i^1;U_i|W,\widetilde{Z}_i^0) = I(W,\widetilde{Z}_i^1;U_i|\widetilde{Z}_i^0)$$
$$= I(W;U_i|\widetilde{Z}_i^0,\widetilde{Z}_i^1) + I(\widetilde{Z}_i^1;U_i|\widetilde{Z}_i^0) = I(W;U_i|\widetilde{Z}_i).$$

$\qquad\square$

**Theorem 3.6** (Restate). *Assume $\ell(\cdot,\cdot) \in [0,1]$, then*

$$d_{\mathrm{JS}}(L_n \,\|\, L_\mu) \leq \frac{1}{n} \sum_{i=1}^{n} I(L_i^0;U_i).$$

*Proof.* By Jensen's inequality and the joint convexity of $d_{\mathrm{KL}}(\cdot \,\|\, \cdot)$, we have

$$d_{\mathrm{KL}}\left(L_n \,\middle\|\, \frac{L_n + L_\mu}{2}\right) = d_{\mathrm{KL}}\left(\frac{1}{n}\sum_{i=1}^{n}\mathbb{E}_{L_i,U_i}\left[L_i^{U_i}\right] \,\middle\|\, \frac{1}{2n}\sum_{i=1}^{n}\mathbb{E}_{L_i}\left[L_i^0 + L_i^1\right]\right)$$

$$\leq \frac{1}{n}\sum_{i=1}^{n} d_{\mathrm{KL}}\left(\mathbb{E}_{L_i,U_i}\left[L_i^{U_i}\right] \,\middle\|\, \frac{1}{2}\mathbb{E}_{L_i}\left[L_i^0 + L_i^1\right]\right) \tag{11}$$

$$= \frac{1}{n}\sum_{i=1}^{n} d_{\mathrm{KL}}\left(\frac{1}{2}\mathbb{E}_{L_i|U_i=0}\left[L_i^0\right] + \frac{1}{2}\mathbb{E}_{L_i|U_i=1}\left[L_i^1\right] \,\middle\|\, \frac{1}{2}\mathbb{E}_{L_i}\left[L_i^0 + L_i^1\right]\right)$$

Similar to Eq. 10, the distributions $P_{L_i^0,U_i}$ and $P_{L_i^1,U_i}$ should satisfy certain symmetry such that

$$P_{L_i^0|U_i=0} = P_{L_i^1|U_i=1}, \quad P_{L_i^1|U_i=0} = P_{L_i^0|U_i=1}, \quad P_{L_i^0} = P_{L_i^1}.$$

Therefore, it satisfies that

$$d_{\mathrm{KL}}\left(L_n \,\middle\|\, \frac{L_n + L_\mu}{2}\right) \leq \frac{1}{n}\sum_{i=1}^{n} d_{\mathrm{KL}}\left(\mathbb{E}_{L_i^0|U_i=0}\left[L_i^0\right] \,\middle\|\, \mathbb{E}_{L_i^0}\left[L_i^0\right]\right).$$

Similarly, we can prove that

$$d_{\mathrm{KL}}\left(L_\mu \,\middle\|\, \frac{L_n + L_\mu}{2}\right) \leq \frac{1}{n}\sum_{i=1}^{n} d_{\mathrm{KL}}\left(\mathbb{E}_{L_i^0|U_i=1}\left[L_i^0\right] \,\middle\|\, \mathbb{E}_{L_i^0}\left[L_i^0\right]\right).$$

Notice that $\frac{1}{2}\left(\mathbb{E}_{L_i^0|U_i=1}\left[L_i^0\right] + \mathbb{E}_{L_i^0|U_i=0}\left[L_i^0\right]\right) = \mathbb{E}_{L_i^0}\left[L_i^0\right]$. Then by combining the two inequalities above and applying Lemma 3.1 with $X = L_i^0$, $Y = U_i$ and $f(L_i^0) = L_i^0$, we have

$$d_{\mathrm{JS}}(L_n \,\|\, L_\mu) = \frac{1}{2}d_{\mathrm{KL}}\left(L_n \,\middle\|\, \frac{L_n + L_\mu}{2}\right) + \frac{1}{2}d_{\mathrm{KL}}\left(L_\mu \,\middle\|\, \frac{L_n + L_\mu}{2}\right)$$

$$\leq \frac{1}{n}\sum_{i=1}^{n} d_{\mathrm{JS}}\left(\mathbb{E}_{L_i^0|U_i=0}\left[L_i^0\right] \,\middle\|\, \mathbb{E}_{L_i^0|U_i=1}\left[L_i^0\right]\right)$$

$$\leq \frac{1}{n}\sum_{i=1}^{n} I(L_i^0;U_i).$$

The proof is complete. $\qquad\square$

**Theorem 3.9** (Restate)**.** *Assume Assumption 3.7 and 3.8 hold and $\ell(\cdot, \cdot) \in \{0, 1\}$, then*

$$L_\mu = d_{\mathrm{JS}}^{-1} \left( L_n, \frac{1}{n} \sum_{i=1}^{n} I(L_i^0; U_i) \right).$$

*Proof.* Notice that Assumption 3.8 implies $\mathbb{E}_{L_i, U_i} \left[ L_i^{U_i} \right] = \mathbb{E}_{L_j, U_j} \left[ L_j^{U_j} \right]$ for any $i, j \in [1, n]$, we have Eq. (11) holds with equality. Therefore,

$$d_{\mathrm{KL}} \left( L_n \, \middle\| \, \frac{L_n + L_\mu}{2} \right) = \frac{1}{n} \sum_{i=1}^{n} d_{\mathrm{KL}} \left( \mathbb{E}_{L_i^0 | U_i = 0} \left[ L_i^0 \right] \, \middle\| \, \mathbb{E}_{L_i^0} \left[ L_i^0 \right] \right),$$

$$d_{\mathrm{KL}} \left( L_\mu \, \middle\| \, \frac{L_n + L_\mu}{2} \right) = \frac{1}{n} \sum_{i=1}^{n} d_{\mathrm{KL}} \left( \mathbb{E}_{L_i^0 | U_i = 1} \left[ L_i^0 \right] \, \middle\| \, \mathbb{E}_{L_i^0} \left[ L_i^0 \right] \right).$$

Then by applying the additional condition of Lemma 3.1 with $X = L_i^0$, $Y = U_i$ and $f(L_i^0) = L_i^0$, we have

$$
\begin{aligned}
d_{\mathrm{JS}}(L_n \, \| \, L_\mu) &= \frac{1}{2n} \sum_{i=1}^{n} d_{\mathrm{KL}} \left( \mathbb{E}_{L_i^0 | U_i = 0} \left[ L_i^0 \right] \, \middle\| \, \mathbb{E}_{L_i^0} \left[ L_i^0 \right] \right) + \frac{1}{2n} \sum_{i=1}^{n} d_{\mathrm{KL}} \left( \mathbb{E}_{L_i^0 | U_i = 1} \left[ L_i^0 \right] \, \middle\| \, \mathbb{E}_{L_i^0} \left[ L_i^0 \right] \right) \\
&= \frac{1}{n} \sum_{i=1}^{n} d_{\mathrm{JS}} \left( \mathbb{E}_{L_i^0 | U_i = 0} \left[ L_i^0 \right] \, \middle\| \, \mathbb{E}_{L_i^0 | U_i = 1} \left[ L_i^0 \right] \right) \\
&= \frac{1}{n} \sum_{i=1}^{n} I(L_i^0; U_i).
\end{aligned}
$$

When Assumption 3.7 holds, by the strong convexity of binary KL divergence, we obtain

$$L_\mu = d_{\mathrm{JS}}^{-1} \left( L_n, \frac{1}{n} \sum_{i=1}^{n} I(L_i^0; U_i) \right),$$

which completes the proof. $\qquad \square$

**Corollary 3.10** (Restate)**.** *Assume Assumption 3.7 and 3.8 hold and $\ell(\cdot, \cdot) \in [0, 1]$, then*

$$L_\mu = d_{\mathrm{JS}}^{-1} \left( L_n, \frac{1}{n} \sum_{i=1}^{n} I(\bar{L}_i^0; U_i) \right).$$

*Proof.* From the definition of the binarized loss $\bar{L}_i^u$, we have $\bar{L}_i^u \in \{0, 1\}$ almost surely and

$$\mathbb{E}_{\bar{L}_i^0 | U_i = 0}[\bar{L}_i^0] = \mathbb{E}_{L_i^0 | U_i = 0}[L_i^0], \qquad \mathbb{E}_{\bar{L}_i^0}[\bar{L}_i^0] = \mathbb{E}_{L_i^0}[L_i^0],$$

for any $i \in [1, n]$ and $u \in \{0, 1\}$. Therefore,

$$
\begin{aligned}
d_{\mathrm{KL}} \left( L_n \, \middle\| \, \frac{L_n + L_\mu}{2} \right) &= \frac{1}{n} \sum_{i=1}^{n} d_{\mathrm{KL}} \left( \mathbb{E}_{L_i^0 | U_i = 0} \left[ L_i^0 \right] \, \middle\| \, \mathbb{E}_{L_i^0} \left[ L_i^0 \right] \right) \\
&= \frac{1}{n} \sum_{i=1}^{n} d_{\mathrm{KL}} \left( \mathbb{E}_{\bar{L}_i^0 | U_i = 0} \left[ \bar{L}_i^0 \right] \, \middle\| \, \mathbb{E}_{\bar{L}_i^0} \left[ \bar{L}_i^0 \right] \right).
\end{aligned}
$$

Similarly, we can prove that

$$d_{\mathrm{KL}} \left( L_\mu \, \middle\| \, \frac{L_n + L_\mu}{2} \right) = \frac{1}{n} \sum_{i=1}^{n} d_{\mathrm{KL}} \left( \mathbb{E}_{\bar{L}_i^0 | U_i = 1} \left[ \bar{L}_i^0 \right] \, \middle\| \, \mathbb{E}_{\bar{L}_i^0} \left[ \bar{L}_i^0 \right] \right).$$

Then by following the proof of Theorem 3.9, we have

$$L_\mu = d_{\mathrm{JS}}^{-1}\left(L_n, \frac{1}{n}\sum_{i=1}^n I(\bar{L}_i^0; U_i)\right).$$

$\square$

**Corollary 3.12** (Restate). *Assume Assumption 3.8 hold and $\delta_j L_n \leq \delta_j L_\mu$ for any $j \geq 1$, then*

$$L_\mu = \sum_{j=1}^\infty d_{\mathrm{JS}}^{-1}\left(\delta_j L_n, \frac{1}{n}\sum_{i=1}^n I(\delta_j \bar{L}_i^0; U_i)\right).$$

*Proof.* From the definition of the truncated loss $\delta_j \bar{L}_i^u$, we have $\delta_j \bar{L}_i^u \in \{0,1\}$ almost surely and

$$\bar{L}_i^u = \sum_{j=1}^\infty \delta_j \bar{L}_i^u,$$

for any $i \in [1, n]$ and $u \in \{0, 1\}$. Then by following the proof of Theorem 3.9, we have

$$d_{\mathrm{JS}}(\delta_j L_n \,\|\, \delta_j L_\mu) = \frac{1}{n}\sum_{i=1}^n I(\delta_j \bar{L}_i^0; U_i),$$

for any $j \geq 1$. Combining with the condition that $\delta_j L_n \leq \delta_j L_\mu$, we finally have

$$L_\mu = \sum_{j=1}^\infty \delta_j L_\mu = \sum_{j=1}^\infty d_{\mathrm{JS}}^{-1}\left(\delta_j L_n, \frac{1}{n}\sum_{i=1}^n I(\delta_j \bar{L}_i^0; U_i)\right).$$

$\square$

**Lemma 4.1** (Restate). *Given random variables $X$, $Y$ such that $Y \sim \mathrm{Bern}\left(\frac{1}{2}\right)$ and $X \in \{0,1\}$ almost surely, then*

$$d_{f\text{-}\mathrm{JS}}\big(\mathbb{E}_{X|Y=0}[X] \,\big\|\, \mathbb{E}_{X|Y=1}[X]\big) = I_f(X; Y).$$

*Proof.* From the definition of $f$-information, we have

$$I_f(X; Y) = \sum_{x\in\{0,1\}}\sum_{y\in\{0,1\}} P(X=x)P(Y=y)f\left(\frac{P(X=x, Y=y)}{P(X=x)P(Y=y)}\right)$$

$$= \frac{1}{2}\sum_{x\in\{0,1\}}\sum_{y\in\{0,1\}} P(X=x)f\left(\frac{P(X=x|Y=y)}{P(X=x)}\right).$$

Noticing that $\frac{\mathbb{E}_{X|Y=0}[X] + \mathbb{E}_{X|Y=1}[X]}{2} = \mathbb{E}_X[X]$, we obtain

$$d_{f\text{-}\mathrm{JS}}\big(\mathbb{E}_{X|Y=0}[X] \,\big\|\, \mathbb{E}_{X|Y=1}[X]\big) = \frac{1}{2}d_f\big(\mathbb{E}_{X|Y=0}[X] \,\big\|\, \mathbb{E}_X[X]\big) + \frac{1}{2}d_f\big(\mathbb{E}_{X|Y=1}[X] \,\big\|\, \mathbb{E}_X[X]\big)$$

$$= \frac{1}{2}\sum_{y\in\{0,1\}} d_f\big(\mathbb{E}_{X|Y=y}[X] \,\big\|\, \mathbb{E}_X[X]\big)$$

$$= \frac{1}{2}\sum_{y\in\{0,1\}} \mathbb{E}_X[X]f\left(\frac{\mathbb{E}_{X|Y=y}[X]}{\mathbb{E}_X[X]}\right) + (1 - \mathbb{E}_X[X])f\left(\frac{1 - \mathbb{E}_{X|Y=y}[X]}{1 - \mathbb{E}_X[X]}\right)$$

$$= \frac{1}{2}\sum_{y\in\{0,1\}} P(X=0)f\left(\frac{P(X=0|Y=y)}{P(X=0)}\right) + P(X=1)f\left(\frac{P(X=1|Y=y)}{P(X=1)}\right)$$

$$= \frac{1}{2}\sum_{x\in\{0,1\}}\sum_{y\in\{0,1\}} P(X=x)f\left(\frac{P(X=x|Y=y)}{P(X=x)}\right).$$

Combining the two equalities above yields the desired result. $\square$

**Theorem 4.2** (Restate). *Assume Assumption 3.8 hold and $\ell(\cdot, \cdot) \in [0, 1]$, then*

$$d_{f\text{-JS}}(L_n \,\|\, L_\mu) = \frac{1}{n} \sum_{i=1}^{n} I_f(\bar{L}_i^0; U_i).$$

*Proof.* By Assumption 3.8 and the definition of $f$-divergence, we have

$$d_f\left(L_n \,\left\|\, \frac{L_n + L_\mu}{2}\right.\right) = d_f\left(\frac{1}{n}\sum_{i=1}^{n} \mathbb{E}_{L_i, U_i}\left[L_i^{U_i}\right] \,\left\|\, \frac{1}{2n}\sum_{i=1}^{n} \mathbb{E}_{L_i}\left[L_i^0 + L_i^1\right]\right.\right)$$

$$= \frac{1}{n}\sum_{i=1}^{n} d_f\left(\mathbb{E}_{L_i, U_i}\left[L_i^{U_i}\right] \,\left\|\, \frac{1}{2}\mathbb{E}_{L_i}\left[L_i^0 + L_i^1\right]\right.\right)$$

$$= \frac{1}{n}\sum_{i=1}^{n} d_f\left(\frac{1}{2}\mathbb{E}_{L_i|U_i=0}\left[L_i^0\right] + \frac{1}{2}\mathbb{E}_{L_i|U_i=1}\left[L_i^1\right] \,\left\|\, \frac{1}{2}\mathbb{E}_{L_i}\left[L_i^0 + L_i^1\right]\right.\right)$$

$$= \frac{1}{n}\sum_{i=1}^{n} d_f\left(\mathbb{E}_{L_i^0|U_i=0}\left[L_i^0\right] \,\left\|\, \mathbb{E}_{L_i^0}\left[L_i^0\right]\right.\right)$$

$$= \frac{1}{n}\sum_{i=1}^{n} d_f\left(\mathbb{E}_{\bar{L}_i^0|U_i=0}\left[\bar{L}_i^0\right] \,\left\|\, \mathbb{E}_{\bar{L}_i^0}\left[\bar{L}_i^0\right]\right.\right).$$

Similarly, we can prove that

$$d_f\left(L_\mu \,\left\|\, \frac{L_n + L_\mu}{2}\right.\right) = \frac{1}{n}\sum_{i=1}^{n} d_f\left(\mathbb{E}_{\bar{L}_i^0|U_i=1}\left[\bar{L}_i^0\right] \,\left\|\, \mathbb{E}_{\bar{L}_i^0}\left[\bar{L}_i^0\right]\right.\right).$$

By applying Lemma 4.1 with $X = \bar{L}_i^0$ and $Y = U_i$, we have

$$d_{f\text{-JS}}(L_n \,\|\, L_\mu) = \frac{1}{2}d_f\left(L_n \,\left\|\, \frac{L_n + L_\mu}{2}\right.\right) + \frac{1}{2}d_f\left(L_\mu \,\left\|\, \frac{L_n + L_\mu}{2}\right.\right)$$

$$= \frac{1}{n}\sum_{i=1}^{n} d_{f\text{-JS}}\left(\mathbb{E}_{\bar{L}_i^0|U_i=0}\left[\bar{L}_i^0\right] \,\left\|\, \mathbb{E}_{\bar{L}_i^0|U_i=1}\left[\bar{L}_i^0\right]\right.\right)$$

$$= \frac{1}{n}\sum_{i=1}^{n} I_f(\bar{L}_i^0; U_i).$$

The proof is complete. $\qquad\square$

**Corollary 4.3** (Restate). *Assume $\ell(\cdot, \cdot) \in [0, 1]$, then*

$$|\overline{\text{gen}}| = \frac{2}{n}\sum_{i=1}^{n} \mathbb{E}_{U_i}\left[\mathbb{W}\left(P_{\bar{L}_i^0|U_i}, P_{\bar{L}_i^0}\right)\right].$$

*Proof.* By taking $f(x) = \frac{1}{2}|x - 1|$, we obtain the total variation metric:

$$d_{\text{TV}}(p \,\|\, q) = |p - q|.$$

We then have

$$d_{\text{TV-JS}}(L_n \,\|\, L_\mu) = \frac{1}{2}d_{\text{TV}}\left(L_n \,\left\|\, \frac{L_n + L_\mu}{2}\right.\right) + \frac{1}{2}d_{\text{TV}}\left(L_\mu \,\left\|\, \frac{L_n + L_\mu}{2}\right.\right)$$

$$= \frac{1}{2}\left|\frac{L_n - L_\mu}{2}\right| + \frac{1}{2}\left|\frac{L_\mu - L_n}{2}\right| = \frac{1}{2}|\overline{\text{gen}}|.$$

By the equivalence between total variation and Wasserstein distance for discrete distributions, we can get

$$
\begin{aligned}
I_{\mathrm{TV}}(\bar{L}_i^0; U_i) &= \sum_{l\{0,1\}} \sum_{u \in \{0,1\}} P(\bar{L}_i^0 = l) P(U_i = u) f\left( \frac{P(\bar{L}_i^0 = l, U_i = u)}{P(\bar{L}_i^0 = l) P(U_i = u)} \right) \\
&= \sum_{u \in \{0,1\}} P(U_i = u) \sum_{l \in \{0,1\}} P(\bar{L}_i^0 = l) f\left( \frac{P(\bar{L}_i^0 = l | U_i = u)}{P(\bar{L}_i^0 = l)} \right) \\
&= \mathbb{E}_{U_i}\left[ D_{\mathrm{TV}}\left( P_{\bar{L}_i^0|U_i} \,\|\, P_{\bar{L}_i^0} \right) \right] \\
&= \mathbb{E}_{U_i}\left[ \mathbb{W}\left( P_{\bar{L}_i^0|U_i}, P_{\bar{L}_i^0} \right) \right].
\end{aligned}
$$

Combining the two equalities above and Theorem 4.2, we obtain

$$
|\overline{\mathrm{gen}}| = 2 d_{\mathrm{TV\text{-}JS}}(L_n \,\|\, L_\mu) = \frac{2}{n} \sum_{i=1}^n I_{\mathrm{TV}}(\bar{L}_i^0; U_i) = \frac{2}{n} \sum_{i=1}^n \mathbb{E}_{U_i}\left[ \mathbb{W}\left( P_{\bar{L}_i^0|U_i}, P_{\bar{L}_i^0} \right) \right].
$$

The proof is complete. $\qquad\square$

## C. Experiment Details and Additional Results

In this section, we present experiment details and additional experimental results that were not included in the main text due to space limitations. The deep learning models are trained with an Intel Xeon CPU (2.10GHz, 48 cores), 256GB memory, and 4 Nvidia Tesla V100 GPUs (32GB). The generalization bounds included in the comparison are listed as follows:

**Theorem C.1.** *(Square-Root), (Theorem 3.2, Wang & Mao (2023a)) Assume $\ell(\cdot, \cdot) \in [0, 1]$, then*

$$
|\overline{\mathrm{gen}}| \le \frac{1}{n} \sum_{i=1}^n \sqrt{2 I(\Delta_i; U_i)}.
$$

**Theorem C.2.** *(Fast-Rate), (Theorem 4.3, Wang & Mao (2023a)) Assume $\ell(\cdot, \cdot) \in [0, 1]$, then for any $C_2 \in \left( 0, \frac{\log 2}{2} \right)$ and $C_1 \ge -\frac{\log(2 - e^{2C_2})}{2C_2} - 1$,*

$$
\overline{\mathrm{gen}} \le C_1 L_n + \frac{1}{nC_2} \sum_{i=1}^n I(L_i^0; U_i).
$$

**Theorem C.3.** *(Binary KL), (Theorem 4.8, Dong et al. (2024a)) Assume $\ell(\cdot, \cdot) \in [0, 1]$, then*

$$
d_{\mathrm{KL}}\left( L_n \,\Big\|\, \frac{L_n + L_\mu}{2} \right) \le \frac{1}{n} \sum_{i=1}^n I(L_i; U_i).
$$

**Theorem C.4.** *(CMI-Oracle), (Theorem 3.1, Wang & Mao (2024)) Assume $\ell(\cdot, \cdot) \in [0, 1]$, then*

$$
|\overline{\mathrm{gen}}| \le \frac{1}{n} \sum_{i=1}^n \sqrt{2 (\mathbb{E}_{\Delta_i}[\Delta_i^2] + |\mathbb{E}_{G_i}[G_i]|) I(\Delta_i; U_i)}.
$$

**Theorem C.5.** *(CSHI-Oracle), (Theorem 3.2, Wang & Mao (2024)) Assume $\ell(\cdot, \cdot) \in [0, 1]$, then*

$$
|\overline{\mathrm{gen}}| \le \frac{1}{n} \sum_{i=1}^n \mathbb{E}_{\widetilde{Z}_i} \sqrt{\left( 4 \mathbb{E}_{\Delta_i|\widetilde{Z}_i}[\Delta_i^2] + 2 |\mathbb{E}_{G_i|\widetilde{Z}_i}[G_i]| \right) I_{\mathrm{H}^2}^{\widetilde{Z}_i}(\Delta_i; U_i)}.
$$

**Theorem C.6.** *(CJSI-Oracle), (Theorem 3.3, Wang & Mao (2024)) Assume $\ell(\cdot, \cdot) \in [0, 1]$, then*

$$
|\overline{\mathrm{gen}}| \le \frac{2}{n} \sum_{i=1}^n \mathbb{E}_{\widetilde{Z}_i} \sqrt{\left( 4 \mathbb{E}_{\Delta_i|\widetilde{Z}_i}[\Delta_i^2] + |\mathbb{E}_{G_i|\widetilde{Z}_i}[G_i]| \right) I_{\mathrm{JS}}^{\widetilde{Z}_i}(\Delta_i; U_i)}.
$$

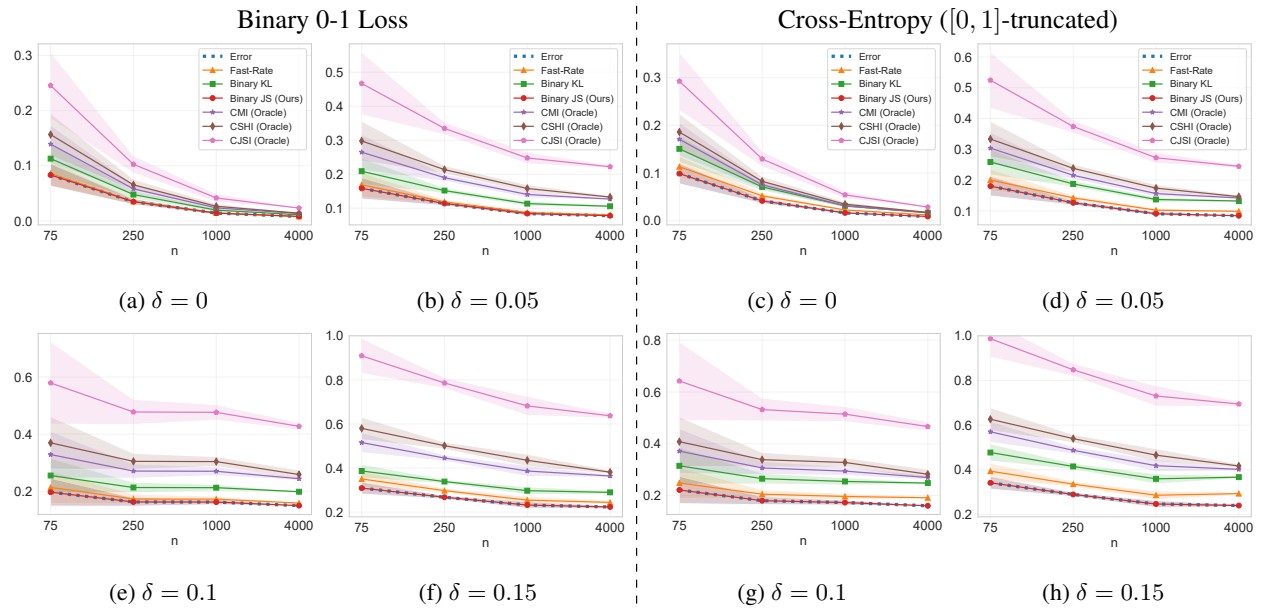

Figure 4: Comparison of generalization bounds under the binary MNIST classification task, where the labels are randomly flipped with probability $\delta$.

In the theorems above, $G_i$ is defined as $G_i \triangleq (-1)^{U_i} \Delta_i$ and satisfies $\mathbb{E}_{G_i}[G_i] = \overline{\text{gen}}$ when Assumption 3.8 holds.

In this experiment, we employ all samples of $(L_i^0, U_i)$, $i \in [1, n]$, simultaneously to estimate the shared joint distribution $P_{L_i^0, U_i}$. This approach differs slightly from prior methods in (Harutyunyan et al., 2021; Hellström & Durisi, 2022b; Wang & Mao, 2023a; 2024), which estimate $P_{L_i^0, U_i}$ using only the samples of the corresponding pair $(L_i^0, U_i)$. Nonetheless, we argue that our method still produces an unbiased estimator for the joint distribution $P_{L_i^0, U_i}$. Taking the binary loss function as an example, let $P_i(l, u) = \mathbb{1}_{L_i^0 = l \wedge U_i = u}$ denote the empirical distribution for sample $(L_i^0, U_i)$. The empirical estimator for $P_{L_i^0, U_i}$ is then given by $P_n = \frac{1}{n} \sum_{i=1}^n P_i(l, u)$. Since $\mathbb{E}_{L_i^0, U_i}[P_i] = P_{L_i^0, U_i}$, it follows that $\mathbb{E}_{L_i^0, U_i}[P_n] = P_{L_i^0, U_i}$, confirming $P_n$ as an unbiased estimator. Therefore, although the samples $(L_i^0, U_i)$, $i \in [1, n]$, are interdependent, our method avoids introducing additional bias in probability density estimation. Furthermore, previous estimators are known to overestimate significantly during the initial stages of SGLD iterations, as noted in (Harutyunyan et al., 2021; Wang & Mao, 2023a). Our approach mitigates this limitation, enabling the estimated upper bounds to accurately track the trends of the generalization error throughout the entire training process.

Our synthetic experimental settings closely follow those in (Wang & Mao, 2024), where synthetic Gaussian datasets are generated using the scikit-learn package. The task involves training a 1-layer linear classification network on 5-dimensional input data points. Class centers are randomly selected from the vertices of a 5-dimensional hypercube, and data instances are then independently drawn from the standard Gaussian distribution at each class center. The model is trained using full-batch gradient descent with a fixed learning rate of 0.01 for 300 epochs. For each test case, we generate 50 distinct supersample datasets $\widetilde{\mathbf{Z}}$, and for each dataset, we draw 100 different supersample variables $U$, resulting in 5,000 independent runs.

In addition, we replicate the experimental settings of (Harutyunyan et al., 2021; Hellström & Durisi, 2022b) for two distinct real-world learning tasks: 1) MNIST (4 vs. 9) classification using a 4-layer CNN network, 2) CIFAR10 classification using a pretrained ResNet-50 network. For each learning task, $k_1$ instances of $\widetilde{\mathbf{Z}}$ are sampled, and for each $\widetilde{\mathbf{Z}}$, $k_2$ samples of $U$ are drawn, yielding $k_1 \times k_2$ independent runs in total. The values of $(k_1, k_2)$ are $(5, 30)$ for MNIST and $(2, 40)$ for CIFAR10, respectively. To further investigate scenarios with substantial overfitting, we conduct an additional experiment introducing random label noise in the binary MNIST classification task. As shown in Figure 4, our Binary JS bound consistently yields exactly tight estimates of the true generalization error across all experiments, including those with significant label noise.

