# OpenReview forum: "Exactly Tight Information-theoretic Generalization Bounds via Binary Jensen-Shannon Divergence"
_ICML.cc/2025/Conference — ICML 2025 poster_

### Official Review · Reviewer_PSQE · 2025-03-13

**Overall Recommendation:** 5

**Summary:**

This paper studies the information-theoretic generalization bounds within the conditional mutual information (CMI) framework by introducing a new information measure called binary Jensen-Shannon (JS) divergence. Specifically, the paper begins with a cleverly designed lemma that builds a relationship between binary JS divergence and mutual information. This key result allows the authors to derive novel, tighter CMI bounds in which the CMI term conditions only on a single random variable. The paper further extends these results by presenting evaluated CMI bounds.
More importantly, under an invariance assumption, the authors demonstrate that the generalization error can be exactly characterized by their binary JS divergence measure. This argument applies not only to the zero-one loss but also to general bounded loss through a novel loss binarization technique. Furthermore, the paper generalizes its findings by extending mutual information and KL-based results to the broader class of $f$-divergence-based results.
Finally, the authors also provide empirical study of their theoretical results, showing that their novel binary JS divergence bounds can exactly characterize generalization, making them tighter than all previous CMI bounds.

**Claims And Evidence:**

All claims are clearly supported.

**Essential References Not Discussed:**

The following paper may need discussion in this work:

[R1] Hellström, Fredrik, and Benjamin Guedj. "Comparing comparators in generalization bounds." International Conference on Artificial Intelligence and Statistics. PMLR, 2024.

The current paper uses binary JS divergence as the comparator between empirical and population loss, whereas [R1] explores a general convex comparator for this purpose and further investigates the optimal convex comparator. Given these conceptual connections, discussing [R1] may provide additional understanding for the choice of binary JS divergence in this work.

**Experimental Designs Or Analyses:**

The experiment settings follow some previous studies and are reasonable to me.

**Methods And Evaluation Criteria:**

The proposed methods and evaluation criteria make sense to me.

**Other Comments Or Suggestions:**

1. In the right column, Lines 110–114, the authors state that "JS divergence serves as a proper metric of distance". This statement is incorrect because **JS divergence does not satisfy the triangle inequality** and therefore is not a proper metric. Please remove this incorrect statement.

2. Please explicitly include Assumption 3.8 in the statements of Theorem 3.9, Corollary 3.10, and Corollary 3.12, as it is crucial for the exact characterization of generalization error in your framework.

3. In the right column, Line 139,  $L=f(W,Z_u)$ should be $L=\ell(W,Z_u)$.

**Other Strengths And Weaknesses:**

Strengths:

1.This paper makes an important technical contribution to the field. While Hellström \& Durisi (2022b) shows that the binary KL term $d(L_n||\frac{L_\mu+L_n}{2})$ is upper bounded by the CMI term, the authors cleverly identify that this binary KL term is embedded within their proposed binary JS measure. More importantly, they utilize the fact that the mutual information between an arbitrary R.V. $X$ and a Bernoulli R.V. $Y$ is equivalent to the JS divergence between $P\_{X|Y=0}$ and $P\_{X|Y=1}$. They further demonstrate that the binary JS divergence between $\mathbb{E}\_{X|Y=0}[X]$ and $\mathbb{E}\_{X|Y=1}[X]$ is exactly equal to this mutual information when $X$ is also binary. This enables an exact characterization of the generalization error using their binary JS measure, and the technique itself may be of independent interest beyond generalization analysis.

2.  The binarization and truncation techniques introduced in this paper are also novel and contribute valuable methodological advancements to the field.

3. Since the binary JS-based bounds provide an exact characterization of generalization error, they overcome key limitations of previous CMI-based bounds, as pointed out in recent works.

Weaknesses:

1. The exact characterization results (e.g., Theorem 3.9, Corollary 3.10) require the algorithm to be invariant to sample permutations. While this is a notable restriction, it is a much more relaxed assumption compared to the interpolating algorithm assumption used in previous works for obtaining exact characterization results.

2. The assumption in Corollary 3.12 may not be easily satisfied in practice. The authors have already acknowledged this limitation in the paper.

**Questions For Authors:**

1. In [R1] (see "Essential References Not Discussed"), it has been shown that when the convex comparator is the Cramér function, one can obtain the tightest possible bound. How do the findings in your paper relate to their results? Could you provide further discussion on the connection between the Cramér function and your binary JS divergence?

2. I thoroughly enjoyed reading this paper, but I have a question regarding the motivation for obtaining the tightest possible CMI bound for generalization error. Specifically, why is it meaningful to derive a generalization bound that is exactly equal to the generalization error? In the CMI setting, where both a training sample and a ghost sample are available, estimating the generalization error directly is feasible. Given this, why should one use your binary JS bound as a predictor of generalization error? Additionally, what unique insights does your exactly tight bound provide, beyond simply computing the generalization error directly in the CMI framework?

**Relation To Broader Scientific Literature:**

This paper is within the literature on learning theory and generalization theory, with a particular focus on the information-theoretic generalization analysis framework. Notably relevant works include Steinke \& Zakynthinou (2020), Hellström \& Durisi (2022b), and Wang \& Mao (2023a).

**Theoretical Claims:**

I checked all the proofs and they seem correct to me.

---

> ### Author Rebuttal · Authors · 2025-03-27
>
> Dear Reviewer PSQE, thank you for your kind words and insightful comments! We address your questions below:
>
> ---
>
> **On the Optimal Convex Comparator**
>
> We appreciate you highlighting this work. As stated in Theorem 4 of that paper, the Cramér function is defined as the convex conjugate of the CGF of a distribution $P_p$ from a set $\mathcal{P}$, where for any $r$ in the loss space $\mathbb{L}$, there exists a $P_r \in \mathcal{P}$ such that $\mathbb{E}[P_r] = r$. This means the Cramér function is not fixed but depends on the loss space.
>
> In their analysis, when $\mathbb{L} = [0,1]$ (as in our setting), $\mathcal{P}$ is the set of all Bernoulli distributions (Eq. (19)), making the Cramér function the binary KL divergence (Eq. (29)). In contrast, our work demonstrates that binary JS divergence outperforms binary KL in the supersample setting, suggesting that their optimality result does not directly extend to our framework.
>
> A key reason may lie in the choice of mutual information measure. Their formulation is based on $D_{\text{KL}}(Q_0 D^n \\| Q_n D^n)$, a generalization of $I(W;\mathbf{Z})$, which reflects the standard hypothesis-based mutual information. In contrast, our analysis focuses on the supersample setting, where the key quantities are $I(W;U|\widetilde{\mathbf{Z}})$ or $I(L;U)$. This may explain from another perspective why our binary JS method (and also the previous fast-rate one) does not apply to this original generalization analysis setting but only the supersample one. Extending their analysis to the supersample setting would be a promising future direction, and we will include these discussions in the revised manuscript.
>
> ---
>
> **On the Value of Tight CMI Bounds**
>
> This is indeed a crucial question for the CMI-based generalization literature. We believe our contribution goes beyond providing tighter estimates to the generalization error. In particular, we address an open question raised by [1]: For which learning problems and learning algorithms is the CMI framework expressive enough to accurately estimate the optimal worst-case generalization error? Our results show that the framework can achieve exact tightness in a broad range of scenarios, offering a conclusive resolution to this line of inquiry.
>
> While our setting is quite general, the practical significance of these bounds becomes more apparent when applied to specific downstream tasks. Due to space limitations, please refer to our response to ```reviewer o7Wh``` for how our results connect to the analysis of out-of-distribution generalization and noisy iterative learning algorithms.
>
> ---
>
> **Other Minor Points**
>
> Thank you for pointing these out. We will make these necessary corrections in the revised version.
>
> ---
>
> [1] Information complexity of stochastic convex optimization: Applications to generalization, memorization, and tracing. ICML, 2024.

---

> > ### Comment · Reviewer_PSQE · 2025-04-03
> >
> > I would like to thank the authors for their responses. Please incorporate the discussion on Hellström and Guedj (2024) into the revised manuscript.
> >
> > Regarding the value of a tight CMI bound, I appreciate the authors' insights. I encourage you to continue reflecting on this in your future work, after all, the tightest possible generalization bound is the generalization error itself. This raises the question: how tight do we truly need a generalization measure to be?
> >
> > Additionally, I would like to point out that an exactly tight IT bound was first given in [1] (rather than Wang and Mao (2023)). I recommend reading Remark 5.5 in the arXiv version (not the ISIT version) of [1], where the authors discuss a similar question.
> >
> > [1] Haghifam, Mahdi, et al. “Understanding Generalization via Leave-One-Out Conditional Mutual Information.” arXiv preprint arXiv:2206.14800 (2022).
> >
> > In light of your responses, I will increase my score.

---

### Official Review · Reviewer_S7W7 · 2025-03-17

**Overall Recommendation:** 3

**Summary:**

This paper investigates the question of tightness in mutual information generalisation bounds. Authors propose exactly tight generalisation bounds based on the binary Jensen-Shannon divergence. They show that their results are also tighter than various existing bounds and successfully involve the impact of a statistical property of optimisation algorithms. They then extend the notion of Jensen-Shannon divergence beyond the KL case and propose associated generalisation bounds. The paper concludes with a numerical assessment of the tightness of their bounds.The

**Claims And Evidence:**

The paper is well-written and pedagogical. The tightness of their results is provably shown, and limitations of the proposed bounds are clearly highlighted.

**Essential References Not Discussed:**

I am not familiar enough with this literature to know whether any crucial reference is missing.

**Experimental Designs Or Analyses:**

The experimental part look sound and coherent with theoretical claims, although I did not check the details.

**Methods And Evaluation Criteria:**

The experimental part consists in simple computations of various generalisation bounds for different pairs (loss, learning algorithms). The  benchmark here consist in the binary KL bound which is a natural comparison point.

**Other Comments Or Suggestions:**

See Questions.

**Other Strengths And Weaknesses:**

See Questions.

**Questions For Authors:**

- In Secs 2.1 and 2.2 you defined twice $L_n$ and $L_\mu$, I assume that it is the definitions of Sec. 2.2 that holds in this work.
- l.134-143 right column: Does this mean that in this work, you always consider $\tilde{Z}_i^0$ to be your training data and $\tilde{Z}_i^1$ the test one?
- More generally, I am not sure to understand the notion of conditioning to $\tilde{Z}_i^0$ in Table 1 when defining your SICIMI framework. If this means that you always assume $\tilde{Z}_i^0$ to be your training data, then is it still relevant to invoke a supersample instead of directly mentioning training and test sets? In this case this would be relevant to discuss the difference with transductive learning, and more particularly the transductive PAC-Bayes learning (see e.g. Begin et al. 2014) which also consider directly a train and test set and propose generalisation bounds involving KL divergences (thus mutual information).
- l.223 right column: would it be possible to briefly describe the proof of convexity of $d_{JS}$?



References:

Begin et al. 2014 PAC-Bayesian Theory for Transductive Learning

**Relation To Broader Scientific Literature:**

I am not familiar enough with this literature to know whether all relevant references have been discussed.

**Theoretical Claims:**

As I am not familiar with literature, I cannot assess the veracity of the proofs. However, the proposed contributions look coherent with existing results, and it is clear to understand the reason of why their results are tighter bounds than existing ones through Figure 2.

---

> ### Author Rebuttal · Authors · 2025-03-27
>
> Dear reviewer S7W7, Thanks for your valuable comments! We are addressing your questions as follows:
>
> ---
>
> **Redefinition of $L_n$ and $L_\mu$**
>
> You are correct that $L_n$ and $L_\mu$ are defined in both Sections 2.1 and 2.2. These two definitions are actually equivalent but expressed using different notations: one for the standard empirical and population risks, and the other adapted to the supersample setting. We will clarify this equivalence in the revised version.
>
> ---
>
> **Training and Testing Separation in $(\widetilde{Z}_i^0, \widetilde{Z}_i^1)$**
>
> The interpretation where $Z_0$ is the training sample and $Z_1$ is the test sample only applies to the illustrative example preceding Section 3.1. In the main analysis, we adopt the supersample setting, where the binary variable $U_i$ determines the training sample $\widetilde{Z}_i^{U_i}$ and the test sample $\widetilde{Z}_i^{\overline{U}_i}$. Hence, $\widetilde{Z}_i^0$ is equally likely to serve as either a training or test sample.
>
> The symmetry in the supersample setting implies that the distributions of $(L_i^0, L_i^1)$ are actually identical under these two procedures:
> - **Original supersample formulation:** Assign training and test samples as $\widetilde{Z}_i^{U_i}$ and $\widetilde{Z}_i^{\overline{U}_i}$ respectively, and define $L_i^0 = \ell(W, \widetilde{Z}_i^0)$, $L_i^1 = \ell(W, \widetilde{Z}_i^1)$.
> - **Illustrative example formulation:** Always fix $\widetilde{Z}_i^0$ for training and $\widetilde{Z}_i^1$ for testing, and define $L_i^0 = \ell(W, \widetilde{Z}_i^{U_i})$, $L_i^1 = \ell(W, \widetilde{Z}_i^{\overline{U}_i})$.
>
> We will revise the paper to unify the example and main analysis settings to avoid confusion.
>
> It is true that our SICIMI term $I(W;U_i|\widetilde{Z}_i^0)$ conditions on $\widetilde{Z}_i^0$, whose distribution reflects a mixture of training and test samples. However, our focus remains on inductive learning algorithms, not transductive ones. Unlike transductive methods, which may leverage unlabeled test data during training, inductive algorithms do not access any information about the test set in the learning phase. The test samples are only used in the analysis stage to evaluate generalization bounds. Therefore, the two setups are fundamentally different and not directly comparable.
>
> ---
>
> **Convexity of $d_{\text{JS}}$**
>
> Here is a proof sketch for this result: The joint convexity of $d_{\text{JS}}$ follows directly from that of the Jensen-Shannon divergence. Specifically, when considering Bernoulli random variables, we have $d_{\text{JS}}(p\\|q) = D_{\text{JS}}(\text{Bern}(p) \\| \text{Bern}(q))$ and $\frac{1}{2} \text{Bern}(p) + \frac{1}{2} \text{Bern}(q) = \text{Bern}\left(\frac{p+q}{2}\right)$. Moreover, the joint convexity of $f$-divergences follows from the convexity of the mapping $(p, q) \mapsto q f(p/q)$, which is inherited from the definition of convex $f$-functions.

---

### Official Review · Reviewer_537o · 2025-03-17

**Overall Recommendation:** 4

**Summary:**

This paper introduces a novel framework for deriving *exactly tight* information-theoretic generalization bounds in machine learning using the binary Jensen-Shannon (JS) divergence. By leveraging a binarization technique for loss variables and supersample frameworks, the authors propose hypothesis-based and prediction-based bounds that address key limitations of prior work, including slow convergence rates and overestimation in deep neural networks. Experiments validate the bounds on synthetic and real-world datasets, demonstrating superiority over baselines like Binary KL divergence and fast-rate bounds.

**Claims And Evidence:**

The claims are all supported by clear and convincing evidence.

**Essential References Not Discussed:**

No

**Experimental Designs Or Analyses:**

The experiment evaluated three different classification tasks, generating Gaussian datasets, 4-layer CNN on binarized MNIST and Pretrained ResNet-50 model on CIFAR10. The experiment is generally reasonable, but it lacks experiments with high-dimensional data, such as ImageNet datasets.

**Methods And Evaluation Criteria:**

The proposed methods  make sense for the problem.

**Other Comments Or Suggestions:**

The related work lacks a discussion and comparison with the literature on PAC-Bayesian generalization bounds, such as
Dupuis, Benjamin, et al. "Uniform Generalization Bounds on Data-Dependent Hypothesis Sets via PAC-Bayesian Theory on Random Sets." Journal of Machine Learning Research 25.409 (2024): 1-55.

https://www.jmlr.org/papers/volume25/24-0605/24-0605.pdf

**Other Strengths And Weaknesses:**

**Strengths**:
1.  This paper proposes a new approach to characterizing the relationship between expected empirical and population risks through a binarized variant of the Jensen-Shannon divergence, which achieves faster convergence compared to existing fast-rate and binary KL-based methods.
2.  Results can be applied to stochastic convex optimization  and extend to f-divergence/Wasserstein metrics .
3. Lemma 3.1 may hold significance beyond the context of generalization analysis,
offering potential applications in broader aspects.

**Weaknesses**:
1. Corollary 3.12 requires $\delta_j L_n \leq \delta_j L_{\mu}$, which seems to be a stronger condition than Assumption 3.7.
2. Lack of experiments on larger datasets.

**Questions For Authors:**

1. This paper introduce  two reasons for producing tight generalization bounds. One is  eliminating redundant random variables from the key mutual information terms (Line 179). Another is using the binary KL divergence (Line 141). Is it necessary to ignore redundant information in conjunction with binary KL divergence to improve the upper bound?
2. Your results rely on ​Assumption 3.7 ($L_n \leq L_{\mu}$). How restrictive are this assumption in practice?

**Relation To Broader Scientific Literature:**

This paper is focus on the theoretical side  and does not have a significant connection with the broader scientific literature.

**Theoretical Claims:**

The paper appears to be technically sound, but I have not carefully checked the details.

---

> ### Author Rebuttal · Authors · 2025-03-27
>
> Dear Reviewer 537o, Thank you for your thoughtful comments and questions! We address them below:
>
> ---
>
> **Assumption in Corollary 3.12**
>
> We agree that the assumption in Corollary 3.12 is stronger than Assumption 3.7. This limitation is acknowledged in Section 5, and we leave the task of relaxing this assumption to future work. Importantly, the core contributions of our paper do not depend on this assumption and are already applicable to a wide range of practical learning scenarios.
>
> ---
>
> **Experiments on Larger Datasets**
>
> While MNIST and CIFAR-10 are relatively simple by today’s standards, they remain common benchmarks in generalization studies (e.g., Harutyunyan et al., 2021; Hellström \& Durisi, 2022b; Wang \& Mao, 2024). Our bounds are designed to scale with arbitrary $n$ and sample distributions, so there is no indication they would deteriorate on larger datasets. ```Reviewer o7Wh``` also found the current experiments sufficient, and we believe they effectively demonstrate our contributions.
>
> ---
>
> **Relation to PAC-Bayesian Bounds**
>
> We agree that PAC-Bayesian bounds are closely connected to information theory, especially through KL divergence terms. However, our work focuses on bounding the *expected* generalization error, whereas PAC-Bayesian approaches typically emphasize *high-probability* bounds. Therefore, the two are not directly comparable. This distinction is also acknowledged by ```Reviewer o7Wh```.
>
> ---
>
> **On Tighter Bounds**
>
> Yes, recent improvements in information-theoretic generalization bounds typically fall into two categories: (1) refining the dependence between mutual information and generalization error—where we propose the binary JS divergence, and (2) improving the information measure itself—where we propose SICIMI for hypothesis-based bounds and bl-MI for prediction-based bounds. These two approaches complement each other to achieve the tightest bounds.
>
> ---
>
> **On the Restrictiveness of Assumption 3.7**
>
> Assumption 3.7 effectively assumes non-negative generalization error, which is typically satisfied by well-trained models. In practice, test performance usually lags behind training performance, and this is exactly the reason why we need generalization analysis. Similar assumptions are also adopted in prior works [1], and it has been shown to always hold for certain algorithms such as Gibbs sampling [2].
>
> ---
>
> [1] Estimation of generalization error: random and fixed inputs. Statistica Sinica, 2006.
>
> [2] An exact characterization of the generalization error for the Gibbs algorithm. NeurIPS, 2021.

---

### Official Review · Reviewer_o7Wh · 2025-03-24

**Overall Recommendation:** 4

**Summary:**

The paper discusses tight information theoretic bounds for the generalization error. The bound is general and can be applied to any machine learning model. This line of work is based on the seminal works of Xu and Raginsky (2017) and follow-up works that use the mutual information between training data and the output of the learning algorithm as a measure of generalization. The research program is motivated by the goal of obtaining a tight computable bound.  Here are the highlights:

* The paper uses the supersample idea (Section 2.2), a data processing like inequality for Jensen-Shannon divergence (Lemma 3.1) to obtain a result.
* Generalization bounds are obtained in Theorem 3.2(where $I(W;U_i|\tilde{Z}^0_i)$ is used) and Theorem 3.6 (where $I(L^0_i;U_i)$ is used).
* Proposition 3.4 shows that the bound based on $I(W;U_i|\tilde{Z}^0_i)$ is strictly tighter.
* In section 3.3, the bound in Theorem 3.6 is shown to be tight for various cases. Finally some extensions are presented in Section 4 based on f-divergence, and the experiments are presented in Section 6.

**Claims And Evidence:**

The main claim is tighter information theoretic bounds backed by proofs, which seem to be sound. The tightness of the bound is shown in various experimental results as well as theoretically proven.

**Essential References Not Discussed:**

Xu and Raginsky themselves cite the original work, where the information-theoretic generalization bound is presented:

Russo and J. Zou, How much does your data exploration overfit? Controlling bias via information usage

It is fair to say that this is the seminal work on the information-theoretic generalization bound, and should be cited.

**Experimental Designs Or Analyses:**

Experiments are conducted for MNIST and CIFAR10 which are considered simple datasets by today’s standard. Nonetheless, for the generalization error analysis, it is sufficient. Many GE works are already vacuous or do not apply for ResNet50 on Cifar10.

The main issue is the low number of training samples used in MNIST and Gaussian experiments.

**Methods And Evaluation Criteria:**

The paper is mainly a theoretical one. The main idea is the use of an inequality based on the Jensen-Shannon divergence and relate that to a mutual information term. The bound is shown to be provably tighter and, in some cases, exactly tight. The key is using the supersample framework and a data processing like inequality for Jensen-Shannon divergence (Lemma 3.1). The key improvement with respect to previous bounds is that the bound is the sum of single samples with the selector random variable in the MI term conditioned on a single sample. I will comment on the utility of these bounds later.

**Other Comments Or Suggestions:**

I wonder if the dependence of $W$ on the training data can be made more explicit for better readability.

**Other Strengths And Weaknesses:**

**Strengths:**

The paper is well written, and the exact tightness is a merit.

**Weakness:**

I would like to clarify a dilemma I have with these information theoretic results. To put it simply, it is not clear what insights these bounds give us about learning. Naively, it seems that these results do not provide any additional insight beyond the fact that the learning algorithm should not memorize the training data, or in this case, it should not memorize the procedure of training data selection.

Besides, the prediction-based generalization bound already involves losses that directly contribute to the precise generalization error, and I wonder whether we are just verifying an algebraic equality, self-fulfilling-ly.

The paper in particular lacks more extensive insights about the results. It presents theorem and plots the numerical results.

It is crucial that the authors clarify what these bounds tell us about learning, how they can be employed in practice, and why machine learning community should care about it. Note that the seminal work of Russo and J. Zou had interesting insights.

**Questions For Authors:**

As I alluded to in my comment above, I wonder if Lemma 3.1 can be obtained using a data processing inequality for f-divergence, knowing that JS-divergence is one (maybe something can be found here f-Divergence Inequalities by Igal Sason, and Sergio Verdu?).

The author mentions the convergence rate of $O(1/n)$ “frequently observed in practical learning scenarios”. Could authors provide additional reference for this claim?

Regarding the bounds in Table 1, including what was presented in the paper: I am not sure if one can read much from the appearance of the term $O(1/n)$ or $O(1/\sqrt{n})$, because the way the other terms scale with $n$ impacts the final dependence. There are many norm-based bounds for deep nets that scale poorly with $n$ despite the explicit $O(1/\sqrt{n})$ dependence. It is difficult to guess the trend from the plots. It might be worth to study the dependence via some curve fitting.

**Relation To Broader Scientific Literature:**

The paper is about the generalization error analysis of learning algorithms. The approach is quite specific and therefore not directly connected to other bounds like PAC Baysian or Rademacher Complexity based bounds, which is fine.

**Theoretical Claims:**

The theoretical claims are properly presented, and the proofs are well-readable and correct. I checked the proofs of Lemma 3.1, Theorem 3.2, and Proposition 3.4 in-depth and looked rapidly at other proofs, which mostly utilize a generally similar proof strategy.

---

> ### Author Rebuttal · Authors · 2025-03-27
>
> Dear reviewer o7Wh, thanks for your thorough reading and constructive questions! We are addressing your questions as follows:
>
> ---
>
> **On the Nature and Significance of Information-Theoretic Results**
>
> This is an insightful and important question involving many works studying information-theoretic bounds. We will clarify the significance of our work from two key perspectives:
>
> **1. Understanding the Limits of the Information-Theoretic Approach**
>
> Recent efforts to tighten bounds have proceeded along two main directions:
> - **Improving dependencies between mutual information and generalization error:** progressing from square-root bounds → binary KL → fast-rate → and now, our binary JS.
> - **Refining the mutual information term itself:** evolving from MI → CMI → $f$-CMI → e-CMI → ld-MI → and finally, our bl-MI.
>
> These efforts have brought increasingly tighter (though still suboptimal) bounds. This naturally raises the question [1]: *For which learning problems is the CMI framework expressive enough to accurately estimate the optimal worst-case generalization error?* Or, are there learning settings where the CMI framework must fail? Some prior works (e.g., Haghifam et al., 2023) explore this on SCO problems. Our work provides a definitive answer: the information-theoretic approach is capable of achieving exactly tight bounds across a broad range of learning scenarios. In doing so, we have adequately addressed this open question and mark a meaningful milestone in this direction.
>
> **2. Strengthening Theoretical Guarantees for Downstream Applications**
>
> It should be noted that our results are developed in a very general setting. They can become especially valuable when applied to more specific contexts. Two particularly promising directions are:
>
> - **Out-of-distribution (OOD) generalization:** Prior works [2,3] use information-theoretic bounds to identify key components for OOD generalization and propose loss-level optimization objectives (e.g., Eq. (5) in [2], Sec. VII.D in [3]). Our results can be adopted to provide a more robust theoretical foundation for such methods.
> - **Understanding noisy, iterative learning algorithms:** For algorithms like SGD and SGLD, information-theoretic bounds (e.g., MI [4], CMI [5]) have been used to analyze the trajectory of the hypothesis and link algorithm behavior with some interesting factors like gradient variance or landscape flatness. We believe our loss-based bounds will further advance this line of works to analyze the loss trajectory (e.g., [6]).
>
> These applications highlight the practical value of our results, though a detailed exploration lies beyond this paper’s scope.
>
> ---
>
> **Alternative Proof for Lemma 3.1**
>
> Thank you for this perspective. While intriguing, we currently do not see a direct derivation of Lemma 3.1 from the $f$-divergence-based data processing inequality. This route may be able to characterize relationships between $d_{\text{JS}}$ and $I_{\text{JS}}$, but not Shannon's mutual information. We consider this a promising direction for future work.
>
> ---
>
> **On the $O(1/n)$ Convergence Rate**
>
> (Strongly) convex optimization is a well-known case exhibiting $O(1/n)$ convergence (e.g., [7]). We agree that information-theoretic bounds with $1/n$ or $\sqrt{1/n}$ terms do not necessarily reflect their real rates. Nevertheless, our aim is not to claim a universal $1/n$ bound, but rather to point out that earlier bounds with explicit $\sqrt{1/n}$ scaling may be inherently suboptimal when faster rates are achievable.
>
> As suggested, we fit the generalization error curves in Figure 3 using $y = ax^b$, and report:
>
> | Dataset  | Gaussian | MNIST | CIFAR-10 |
> |----------|----------|-------|----------|
> | $b$      | -1.084   | -0.585| -0.326   |
>
> This indicates a convergence rate near $O(1/n)$ for synthetic data and closer to $O(\sqrt{1/n})$ on real datasets.
>
> ---
>
> **Other Points**
>
> We now cite Russo and Zou’s seminal work, and explicitly denote $W = \mathcal{A}(\mathbf{Z})$ to clarify the dependence on training data. Regarding the number of training samples, we note that the curves for different bounds already converge closely at large $n$, and thus increasing $n$ may not further enhance this comparison.
>
> ---
>
> [1] Information complexity of stochastic convex optimization: Applications to generalization, memorization, and tracing. ICML, 2024.
>
> [2] On $f$-Divergence Principled Domain Adaptation: An Improved Framework. NeurIPS, 2024.
>
> [3] How Does Distribution Matching Help Domain Generalization: An Information-theoretic Analysis. TIT, 2025.
>
> [4] On the generalization of models trained with SGD: Information-theoretic bounds and implications. ICLR, 2022.
>
> [5] Sharpened generalization bounds based on conditional mutual information and an application to noisy, iterative algorithms. NeurIPS, 2020.
>
> [6] Analyzing generalization of neural networks through loss path kernels. NeurIPS, 2023.
>
> [7] Train faster, generalize better: Stability of stochastic gradient descent. ICML, 2016.

---

> > ### Comment · Reviewer_o7Wh · 2025-04-02
> >
> > I would like to thank the authors for their answers. Particularly, thanks for the comments on the convergence rate. I suggest to include this discussion in the final version and clarify these subtleties.
> >
> > Regarding your answer to \textit{Significance of Information-Theoretic Results}, the authors have tried to clarify further their contribution in the first point. I am not questioning this, and I acknowledge the progress made in this paper. However, my question was more general: \textit{what are we learning from these bounds? how can they impact the machine learning research and practice?}  The examples provided in the second bullet point provide promising directions to address this question. However, I cannot think of similar examples with concrete outcome in the previous literature on IT generation bounds, so I tend to think that the lack of application is a weakness of this framework.
> >
> > Overall, I think the paper clearly passes the bar for acceptance, so I change my score to reflect that.

---

### Decision · Program_Chairs · 2025-05-01

**Decision:**

Accept (poster)

**Comment:**

The paper discusses tight information theoretic bounds for the generalization error. The bound is general and can be applied to any machine learning model. This line of work is based on the seminal works of Xu and Raginsky (2017) and follow-up works that use the mutual information between training data and the output of the learning algorithm as a measure of generalization. The research program is motivated by the goal of obtaining a tight computable bound.